# A clinical transcriptome approach to patient stratification and therapy selection in acute myeloid leukemia

T. Roderick Docking [1,2], Jeremy D. K. Parker [2], Martin Jädersten[2], Gerben Duns[2], Linda Chang[2], Jihong Jiang[2], Jessica A. Pilsworth[2], Lucas A. Swanson [2], Simon K. Chan[2], Readman Chiu[2], Ka Ming Nip [2], Samantha Mar[2], Angela Mo[2], Xuan Wang[2], Sergio Martinez-Høyer[2], Ryan J. Stubbins[2,3], Karen L. Mungall[2], Andrew J. Mungall [2], Richard A. Moore[2], Steven J. M. Jones [2,4], İnanç Birol [2,4,5], Marco A. Marra [2,4], Donna Hogge[6] & Aly Karsan [1,2,5 ✉]

As more clinically-relevant genomic features of myeloid malignancies are revealed, it has become clear that targeted clinical genetic testing is inadequate for risk stratification. Here, we develop and validate a clinical transcriptome-based assay for stratification of acute myeloid leukemia (AML). Comparison of ribonucleic acid sequencing (RNA-Seq) to whole genome and exome sequencing reveals that a standalone RNA-Seq assay offers the greatest diagnostic return, enabling identification of expressed gene fusions, single nucleotide and short insertion/deletion variants, and whole-transcriptome expression information. Expression data from 154 AML patients are used to develop a novel AML prognostic score, which is strongly associated with patient outcomes across 620 patients from three independent cohorts, and 42 patients from a prospective cohort. When combined with molecular risk guidelines, the risk score allows for the re-stratification of 22.1 to 25.3% of AML patients from three independent cohorts into correct risk groups. Within the adverse-risk subgroup, we identify a subset of patients characterized by dysregulated integrin signaling and *RUNX1* or *TP53* mutation. We show that these patients may benefit from therapy with inhibitors of focal adhesion kinase, encoded by *PTK2*, demonstrating additional utility of transcriptome-based testing for therapy selection in myeloid malignancy.

[1] Experimental Medicine Program, Department of Medicine, University of British Columbia, Vancouver, BC, Canada. [2] Canada's Michael Smith Genome Sciences Centre, BC Cancer, Vancouver, BC, Canada. [3] Department of Medicine, University of British Columbia, Vancouver, BC, Canada. [4] Department of Medical Genetics, University of British Columbia, Vancouver, BC, Canada. [5] Department of Pathology and Laboratory Medicine, University of British Columbia, Vancouver, BC, Canada. [6] Leukemia Bone Marrow Transplant Program of BC, Vancouver General Hospital, Vancouver, BC, Canada. ✉email: akarsan@bcgsc.ca

The myeloid malignancies consist of a group of related hematopoietic stem/progenitor cell cancers, including acute myeloid leukemia (AML) and myelodysplastic syndromes (MDS)[1–3]. The prognosis for AML patients older than sixty years of age has not substantially improved in decades and remains dismal. This lack of progress highlights the need for improved diagnostic approaches, clinical assessment, and treatment strategies, all of which require a better understanding of the genetic basis of these diseases. Currently, clinical diagnostics and risk stratification for AML rely on cytogenetic screening for structural genomic alterations and targeted sequence-based screening for prognostic and predictive genetic variants[1,4]. However, ~50% of patients are stratified to an intermediate-risk group and remain difficult to assign to an appropriate consolidation of therapy regimen[1], exemplifying the need for improved stratification of AML patients.

While standard-of-care clinical guidelines used for AML stratification are relatively conservative in their use of novel genetic markers, several recent studies have proposed revised stratification schemes. Recent updates to the European LeukemiaNet (ELN) guidelines for AML stratification include *RUNX1*, *TP53*, and *ASXL1*[5], and more recently published schemes incorporate mutational status in 25 genes and propose 14 separate disease subtypes, although this has not been clinically incorporated[6]. Additional models using mutational and clinical data[7] or gene expression profiling[8–10] have also been proposed for improved patient stratification.

While next-generation sequencing of small gene panels is currently common for diagnostic purposes, genome-wide screening offers many potential benefits[11]. RNA-Seq assays are typically treated as complementary to existing DNA-based assays rather than as stand-alone assays. In some cases, however, RNA-Seq has been shown to be more useful than whole-exome sequencing (WES) or whole-genome sequencing (WGS) in providing actionable clinical hypotheses in cancer[12,13] and Mendelian disorders[14,15]. RNA-Seq assays offer potential clinical benefits including the ability to detect expressed structural variants (SVs), alternative isoform usage and splicing variation, and global gene expression[16], all of which are known to be relevant for understanding the pathogenesis of myeloid malignancies. In addition, RNA-Seq has the potential to supplant cytogenetic testing, potentially improving cost-effectiveness. However, for RNA-Seq assays to be incorporated into clinical workflows, strict standards for analytic and clinical validity must be demonstrated and a quality framework established[17,18].

In this work, we show that RNA-Seq based testing exceeds the current clinical standard of care for the assessment of myeloid malignancies, and provides the broadest range of genomic information, when compared to WES- and WGS-based approaches. Further, we develop a novel gene expression signature that allows for the restratification of cases classified by the current protocol as intermediate-risk AMLs into high- or low-risk subgroups, thereby allowing better risk stratification for clinical management. Finally, to demonstrate the utility of transcriptome-based testing in improving therapy selection in AML, we identify a subset of high-risk patients with dysregulated integrin signaling, which is potentially amenable to inhibitors of focal adhesion kinase (FAK). As therapeutic options for myeloid malignancies continue to evolve, a global transcriptome-based approach to diagnostics will allow reconfiguration of mutation- and expression-based predictors to best take advantage of new genomic information as it arises.

## Results

**Experimental design and quality control.** To compare RNA-Seq, WGS, and WES as potential platforms for clinical assessment of myeloid malignancies, we constructed a patient cohort (the AML Personalized Medicine Program, or AML PMP cohort) consisting of patients with de novo AML, secondary AML (sAML), therapy-related AML (tAML), MDS, and therapy-related MDS (tMDS). To demonstrate the analytic validity of the RNA-Seq pipeline, we constructed a separate validation cohort consisting of replicated patient and cell line material, analyzed a local prospective cohort of newly diagnosed patients, and re-analyzed patient RNA-Seq libraries from The Cancer Genome Atlas AML (TCGA LAML)[19], Beat AML[20], and TARGET AML[21] cohorts (Fig. 1A, Supplementary Data 1–8, Supplementary Fig. 1). We generated and compared quality metrics for the AML PMP cohort, determined appropriate quality thresholds (Supplementary Fig. 2), and compared coverage depth across sequencing platforms (Supplementary Fig. 3) for a set of 44 genes with established disease relevance (Supplementary Data 9).

**SNV and small indel detection.** SNVs and short indel detection can be problematic in RNA-based assays due to varying levels of transcription, the difficulty of spliced alignment, allele-specific expression, and RNA editing[16]. We therefore assessed concordance, sequence coverage, and variant allele frequency (VAF) in matched RNA-Seq, WES, and WGS libraries, using three different variant calling algorithms (GATK HaplotypeCaller[22], FreeBayes[23], and VarScan2)[24] and an ensemble caller. Variant coverage depth was considerably higher for a given RNA-Seq variant (median coverage: 317 and 168 across the RNA-Seq cohorts) compared to WES (median coverage: 112) or WGS (median coverage: 36) variants (Supplementary Fig. 4, Supplementary Data 10). These observed coverage depths imply that sequencing RNA-Seq libraries with fewer total reads would still provide sufficient sequencing coverage for high-confidence variant detection (e.g., sequencing ~60 million reads per library would result in an expected mean variant depth of 100x), although deeper sequencing would be expected to improve variant detection sensitivity.

The majority of variants were concordant between matched RNA-Seq and WGS (84/94 concordant calls) or WES (375/402 concordant calls) libraries. Comparison of RNA-Seq to WGS revealed eight cases where variant calls were present in the RNA-Seq but not WGS data, and two variants in the WGS data that were not observed in the matched RNA-Seq libraries (Fig. 1B, Supplementary Data 11). Comparison of RNA-Seq to WES revealed 19 calls present in RNA-Seq but not WES data, and eight calls unique to the WES libraries (Fig. 1C, Supplementary Data 12). Seven out of eight of the variants that were unique to WES and were not found in RNA-Seq data were rare X-linked variants (in *BCORL1*, *BCOR*, and *SMC1A*) in female patients, suggesting the variant alleles reside on the inactivated X chromosome (Supplementary Data 13). These variants are all annotated as benign, of unknown significance, or unreported in ClinVar. For the variants of unknown significance, their lack of expression at the transcript level suggests they are likely benign changes. The eighth WES-specific variant was located on chromosome 3 (*GATA2* p.Pro161Ala), and is annotated as a likely benign germline change in ClinVar. In general, we observed biased expression for variants in *WT1* and *GATA2*, which are located on chromosomes 11 and 3, respectively, and are known to be imprinted[25,26]. Variants present in RNA-Seq libraries but not in matched WES or WGS libraries were mostly discordant due to low coverage in the matched WES or WGS data. However, several cases where variants were present at low allelic frequencies in the WES and WGS data were found to be highly expressed in the RNA-Seq data. These variants included a nonsense mutation in *WT1*, and a *KIT* p.Asp816Tyr change, both with prognostic relevance that would be likely missed in WGS or WES data.

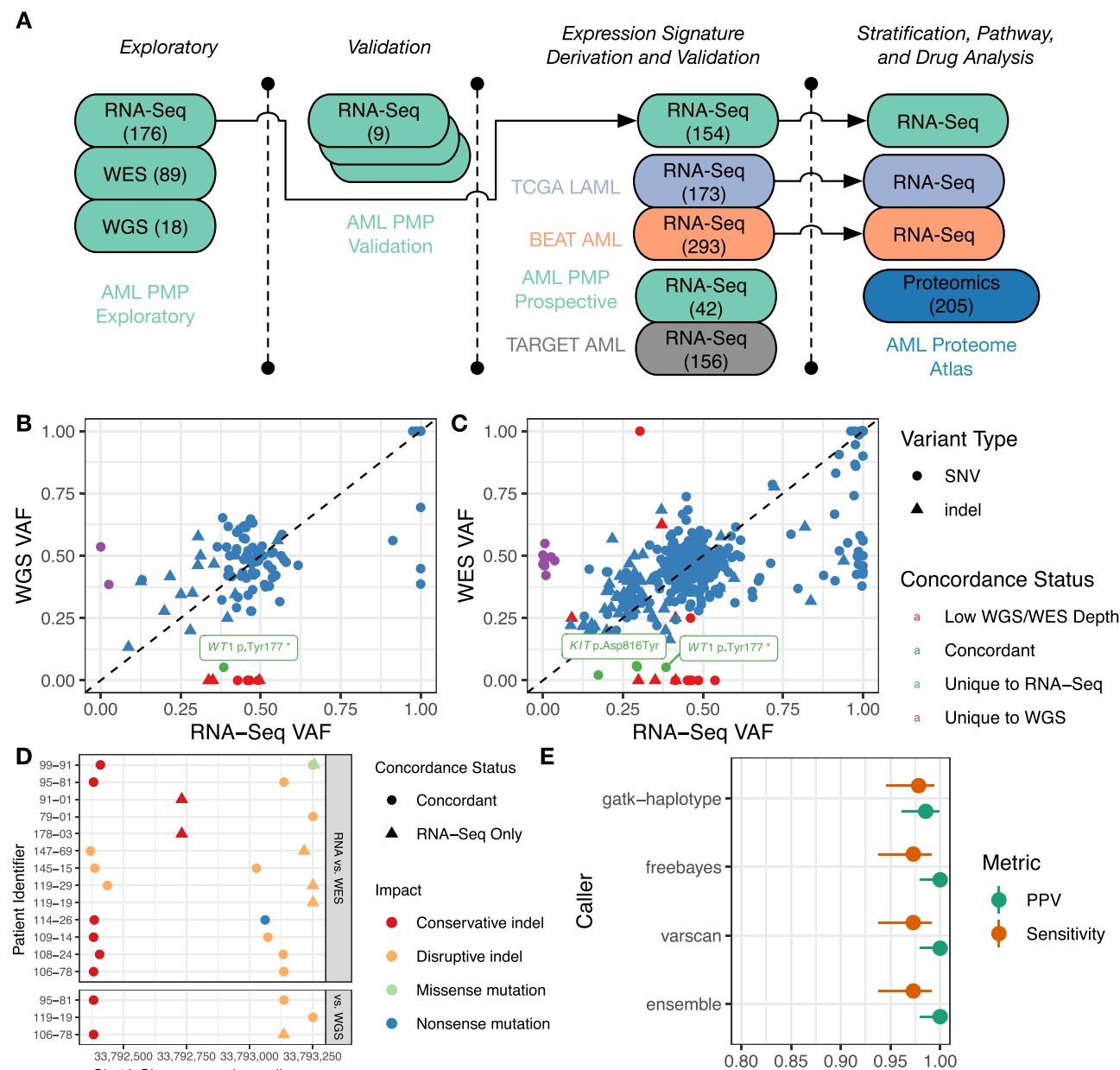

**Fig. 1 Experimental overview and short nucleotide variant/indel analysis. A** Overview of datasets used at each stage of the analysis, colored by sequencing project, with sample size for each data set indicated in brackets. For the AML PMP exploratory batch, MDS samples were used for profiling the SNV and gene fusion pipelines, but not for expression-based analyses, and validation RNA-Seq libraries were prepared in triplicate. **B–C** Matched variant allele frequency (VAF) for variants between whole-genome sequencing (WGS) and RNA-Seq (**B**) and whole exome sequencing (WES) and RNA-Seq (**C**), by concordance status, variant type, and coverage status (sites with ≤10x coverage are indicated as 'Low WGS/WES Depth'). Selected variants discussed in the text are labeled with Human Genome Variation Society (HGVS) nomenclature. **D** Variant observations in *CEBPA*. Potentially disruptive mutations for each patient (*y*-axis) are indicated by their chromosomal coordinate (*x*-axis), with predicted impact and variant concordance. **E** Sensitivity and positive predictive value (PPV) for each variant caller in the validation cohort (with 95% confidence intervals).

Proper assignment of mono-allelic or bi-allelic mutation status in *CEBPA* is necessary for patient stratification[27]. Sequence coverage across this GC-rich gene was particularly poor in WES libraries, leading to six additional calls being recovered in the RNA-Seq data compared to WES (Fig. 1D, Supplementary Fig. 5). Comparison of RNA-Seq to WGS revealed a single *CEBPA* variant that was missed in the WGS data.

**Validation analysis.** To characterize the reproducibility of the RNA-Seq variant-calling pipeline, we compared filtered call sets from technical replicate samples to curated reference sets (Supplementary Figs. 6–7, Supplementary Data 5, 14–15), calculating the sensitivity and positive predictive value (PPV) of the observed variants for each individual variant caller, as well as the ensemble caller. Of the single callers, GATK Haplotype-Caller had the highest sensitivity (0.978 (0.946–0.994)) and lowest PPV (0.985 (0.961–0.999)) (Fig. 1E). The ensemble caller improved on the PPV of GATK HaplotypeCaller (1 (0.98–1)), but not the sensitivity (0.973 (0.938–0.991)). Notably, the difference in sensitivities between callers was solely due to GATK HaplotypeCaller recovering two additional synonymous variants compared to the other calls (while at the same

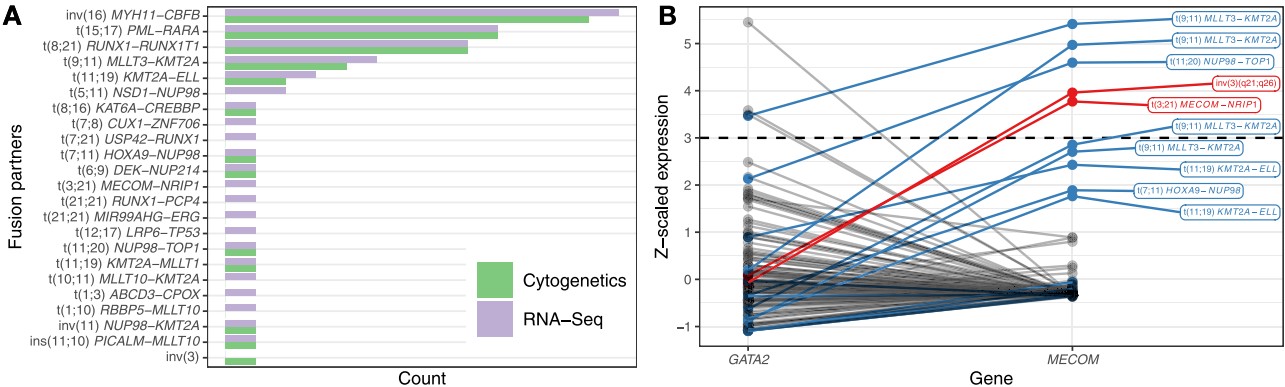

**Fig. 2 Structural variation analysis. A** Number of structural variation events detected by cytogenetics analysis vs. RNA-Seq fusion detection. **B** z-scaled gene expression of *GATA2* and *MECOM* in the AML PMP exploratory patient cohort. *MECOM*- (red) and *KMT2A*-related (blue) structural variants are colored. The dashed horizontal line indicates the threshold of $z \geq 3$ for *MECOM* outlier expression.

time not recovering a third synonymous variant) (Supplementary Data 16). The remaining false-negative calls consisted of the same X-linked synonymous variants observed in the exploratory cohort (Supplementary Data 17), while the GATK false-positive calls consisted of rare alignment artefacts around splice junctions (Supplementary Data 18). While all three callers showed similar high performance across the target space, we chose to use GATK HaplotypeCaller for subsequent experiments in order to simplify downstream analyses.

**Structural variation.** The myeloid malignancies are characterized by the presence of recurrent SVs. We observed complete sensitivity in the exploratory cohort of 173 patients with RNA-Seq libraries for detection of the main clinically relevant gene fusions, including *PML-RARA* (nine cases), *MYH11-CBFB* (11 cases), *RUNX1-RUNX1T1* (eight cases), and *MLLT3-KMT2A* (four cases) (Fig. 2A), in addition to 11 rarer translocations (Supplementary Data 19). After establishing filtering criteria for novel fusions, we observed several interesting novel fusion events which either recapitulated the known cytogenetic events in greater resolution, or identified cryptic events which were not detected by cytogenetics (Supplementary Fig. 8A, Supplementary Data 20–22). For example, we observed multiple *CUX1* fusions[28] in a patient with sAML, complex karyotype, and *TP53* mutations. Similarly, we observed an *ERG* fusion[29] in another patient with sAML, complex karyotype, and *TP53* mutation. After review, we identified a total of 18 novel gene fusions in disease-related genes after filtering, including seven patients with novel rearrangements in the *KMT2A* gene family (Supplementary Fig. 8B). These novel fusions would likely have altered treatment decisions in several cases.

The only case where a clinically relevant chromosomal rearrangement was detected by cytogenetics, but was not detected by RNA-Seq was a *MECOM* (*EVI1*)-*RPN1* fusion representing an inv(3) karyotypic rearrangement. This case was not recovered since the fusion breakpoint occurs downstream of both genes[30]. Since inv(3) rearrangements induce aberrant expression of *MECOM* by relocating a *GATA2* enhancer sequence, we compared the relative expression of these genes (Fig. 2B). The relevant case exhibited the expected pattern of elevated *MECOM* expression, as did another *MECOM*-rearranged sample. High *MECOM* expression was also observed in *KMT2A*-family rearranged AMLs, as noted previously[31,32] (Supplementary Data 23). A similar pattern was observed in the TCGA LAML cohort (Supplementary Data 24). These findings indicate that *MECOM* rearrangements can be detected using expression analysis to

identify cases with high *MECOM* expression, but which lack *KMT2A*-family rearrangements.

We further evaluated the accuracy of our pipeline for two other clinically relevant SVs: internal tandem duplications (ITDs) in *FLT3* (Supplementary Figs. 8C, D) and partial tandem duplications (PTD) in *KMT2A* (*MLL*). We observed complete analytic sensitivity (33/33 positive results recovered) for *FLT3*-ITD detection, and also observed ten novel *FLT3*-ITD calls in cases where a prior PCR-based assay showed a negative result. These novel events were validated by panel sequencing (eight cases) or manual review (two cases), and consisted mainly of events with very low estimated burdens (Supplementary Fig. 9). Although low- and high-burden *FLT3*-ITDs are thought to confer differing risk status[33], we did not observe a difference in outcomes for patients between low and high allele fraction *FLT3*-ITD variants (Supplementary Fig. 10). We also observed ten *KMT2A* (*MLL*) PTD events, which are not detectable by standard clinical assays, but are crucial for proper assessment of patient risk status[34]. In five of these cases, detection of the PTD event resulted in a change of patient stratification from intermediate to adverse by ELN guidelines[5].

**Gene expression signature—AML prognostic score (APS).** Having validated the RNA-Seq assay as an improvement upon current clinical assay performance, we next sought to derive a gene expression signature that could be used to improve risk stratification for a broad spectrum of AMLs. We used least absolute shrinkage and selection operator (LASSO) regression[35] to derive a 16-gene expression signature, which we named the APS (Fig. 3A, Supplementary Data 25). This model included both genes with previously described leukemic associations, such as *CD109*[36], and genes with previously undescribed leukemic associations. The APS value was strongly associated with overall survival in the AML PMP exploratory cohort (HR = 5.46, p = 4.57e−13, Fig. 3B). To validate the utility of the gene expression signature, we applied the APS to two separate validation cohorts, observing a strong association with overall survival in both the TCGA LAML (HR = 2.52, p = 2.48e−06, Fig. 3C) and Beat AML (HR = 2.43, p = 1.88e−06, Fig. 3D) cohorts. We then tested the utility of the APS signature in a prospective local cohort, again observing a strong association with overall survival (HR = 2.97, p = 0.00676, Fig. 3E). Interestingly, APS was also predictive in a pediatric AML cohort[21] (HR = 2.16, p = 0.00115, Fig. 3F). The APS value also appears robust to changes in library preparation protocol, as matched libraries prepared using ribo-depletion protocols showed well-correlated gene expression values for the

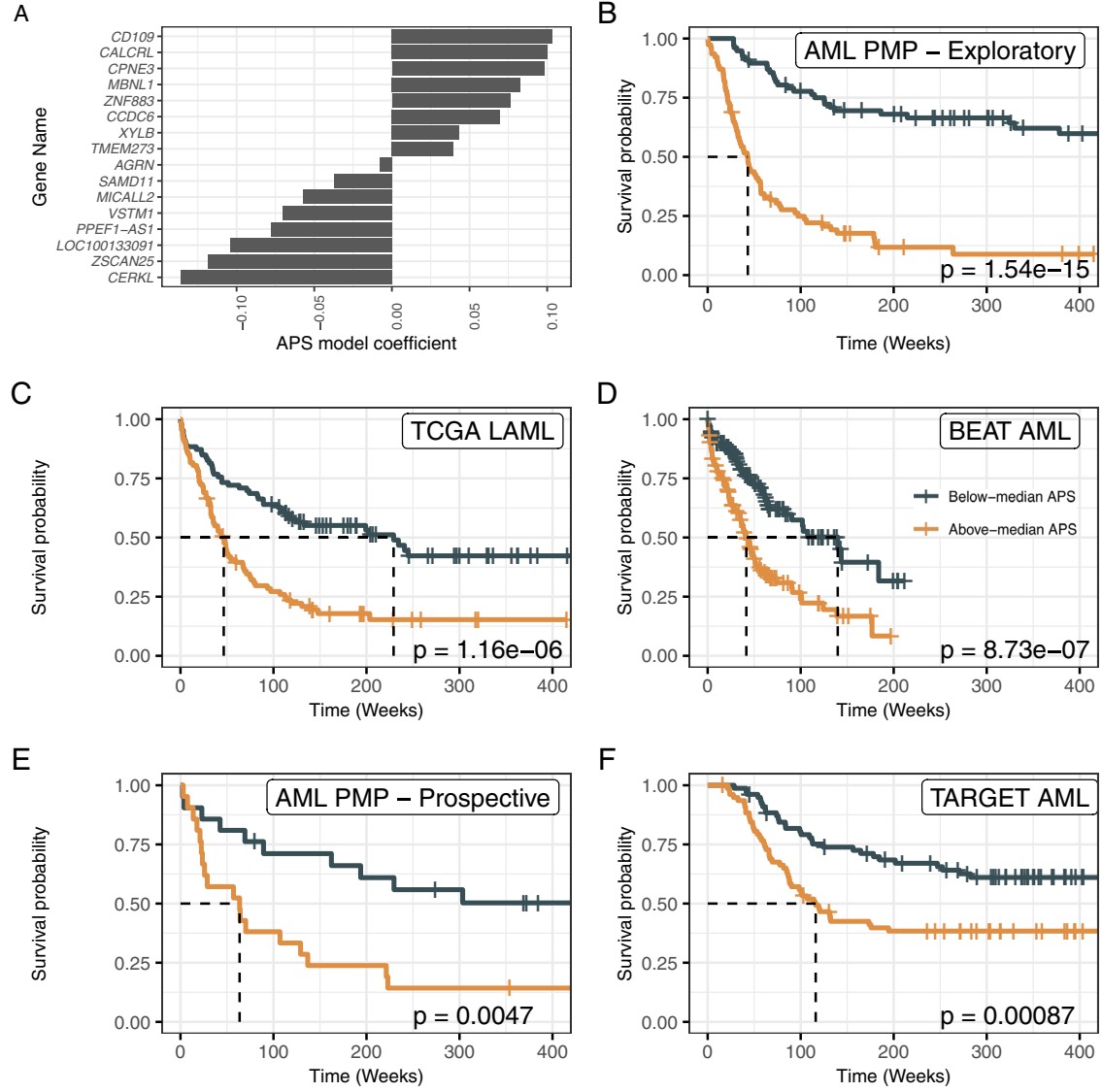

**Fig. 3 Training and validation of the AML prognostic score (APS) gene expression signature. A** APS model coefficients. The *y*-axis indicates the genes making up the APS set, with the *x*-axis indicating the model coefficients. **B**–**F** Survival plots for the AML PMP (**B**, *n* = 154), TCGA LAML (**C**, *n* = 173), Beat AML (**D**, *n* = 293), AML PMP Prospective (**E**, *n* = 42), and TARGET pediatric AML (**F**, *n* = 156) cohorts, for above-median and below-median values of the APS value within each cohort. Dashed lines indicate time to median survival. Log-rank *p* values are indicated for each cohort.

signature genes (for ribo-depleted libraries with good mapping rates, Supplementary Fig. 11).

It has previously been demonstrated that the LSC17 score, based on the expression of 17 genes linked to leukemic stem cells (LSCs), can be used to identify AML cases with poor prognosis and treatment resistance[10]. We calculated LSC17 scores for each of the de novo AML (including favorable-risk cases), tAML, and sAML samples, and observed that samples with above-median scores (within each cohort) had significantly worse survival outcomes in the AML PMP (HR = 1.99, *p* = 0.000995), TCGA LAML (HR = 1.86, *p* = 0.00116), and Beat AML (HR = 1.69, *p* = 0.00324) cohorts. However, in each case, the hazard ratio for the APS model was higher than the LSC17 model. In comparing the APS values and LSC17 scores, we observed that most patients had the same high/low status for both markers (AML PMP: 66.2%, TCGA LAML: 69.9%, Beat AML: 63.8%, Supplementary Fig. 12, Supplementary Data 26). In multivariate survival models including ELN risk category and either the APS value or LSC17 score, we saw that the APS value was associated with a

larger hazard ratio and stronger significance in both the AML PMP (APS: HR = 3.6, *p* = 5.76e-06, LSC17: HR = 1.63, *p* = 0.0235) and TCGA LAML (APS: HR = 1.98, *p* = 0.00567, LSC17: HR = 1.46, *p* = 0.0582) cohorts (Supplementary Fig. 13).

**Expression-based stratification.** To determine whether the addition of gene expression information could improve AML risk stratification, we analyzed the contributions of clinical and molecular predictors in the AML PMP exploratory cohort using univariate survival analysis (Fig. 4A, Supplementary Data 27). Older patients (age ≥ 60 at diagnosis) had very poor outcomes (HR = 3.11, adj. *p* = 1.32e−05), while the remaining clinical variables were non-significant. We observed a strong association between the presence of a *KMT2A*-related fusion or PTD with poor outcomes (HR = 3.71, adj. *p* = 6.13e−05), and common molecular alterations such as *NPM1* and *FLT3*-ITD showed the expected positive and negative associations with outcome[5]. *TP53* mutations (HR = 4.59, adj. *p* = 0.00163) showed a strongly adverse effect as previously reported[37].

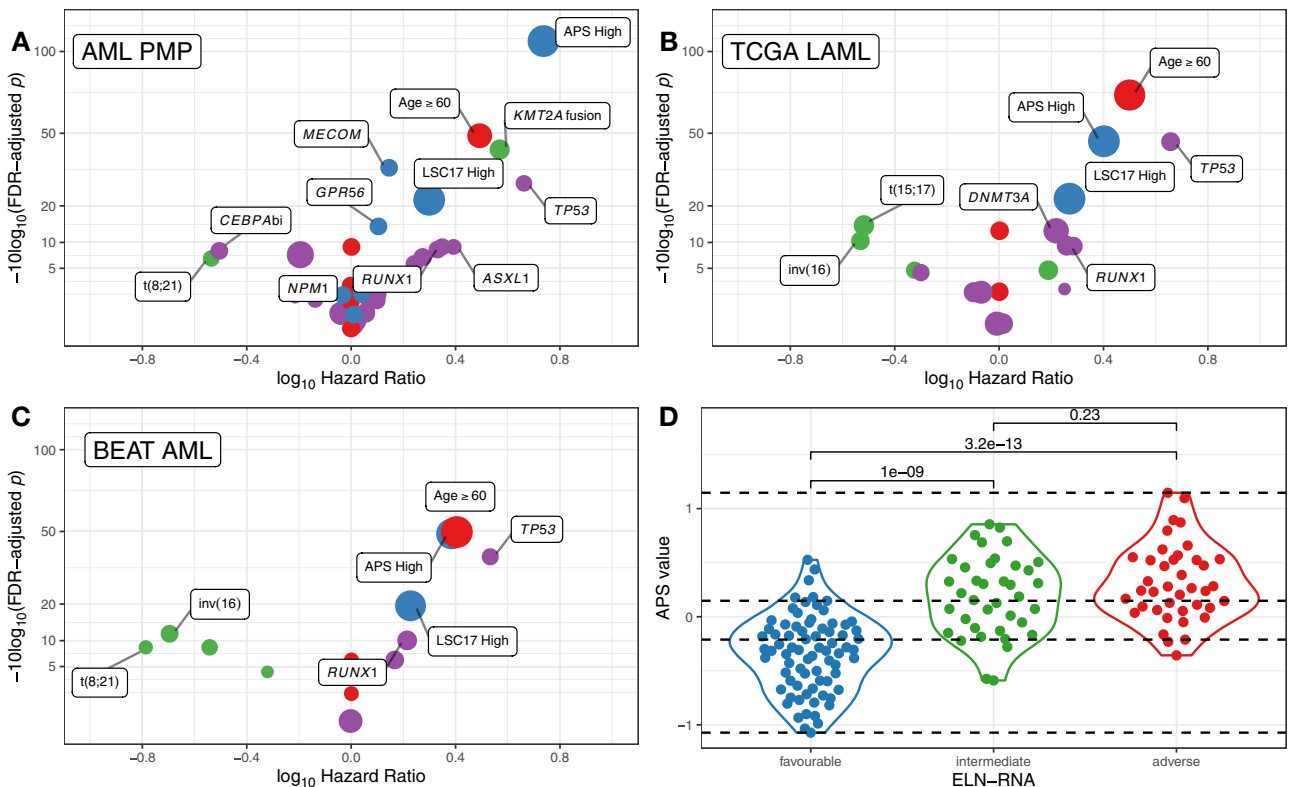

**Fig. 4 Univariate survival analysis. A–C** Univariate survival analysis for selected clinical, gene fusion, mutation, stratification, and expression-based predictors of survival for the AML PMP (**A**), TCGA LAML (**B**), and Beat AML (**C**) cohorts. In each panel, the *x*-axis indicates the hazard ratio, and the *y*-axis indicates the FDR-adjusted *p*-value for each tested variable for univariate Cox proportional hazards models. Point sizes are scaled to reflect the proportion of patients affected by the relevant marker, and colored as clinical variables—red, gene fusions—green, SNV/indels—purple, expression—blue. **D** Distribution of APS values for AML PMP patients, colored by ELN-RNA risk status (favorable —blue, intermediate—green, adverse—red). Horizontal dashed lines indicate the bottom, intermediate, and top terciles of the APS value distribution. Comparison bars show two-sided Wilcoxon test *p*-values for each pairwise comparison.

The expression level of several genes has been previously described as having prognostic relevance for AML, including *GPR56*[38], *BAALC*[39], *MN1*[40], *MECOM*[8,31], and *FLT3*[13]. We included continuous *z*-scaled expression of these genes (Supplementary Fig. 14), in addition to the APS and LSC17 multigene expression signatures in the univariate analysis. Patients with above-median APS values had the highest observed hazard ratio (HR = 5.46, adj. *p* = 1.74e−11). We also observed significant effects of above-median LSC17 expression (HR = 1.99, adj. *p* = 0.0063), *MECOM* expression (HR = 1.4, adj. *p* = 0.000392), and *GPR56* expression (HR = 1.27, adj. *p* = 0.0401). We observed similar results in both the TCGA LAML (Fig. 4B) and Beat AML cohorts (Fig. 4C). In these validation cohorts, the highest hazard ratios were associated with age, *TP53* mutations, and high APS values.

To re-stratify the patient cohorts, we first used ELN criteria[5] deriving information only from the RNA-Seq assay (ELN-RNA stratification, Supplementary Fig. 15, Supplementary Data 28) to stratify the patient cohorts into favorable-, intermediate-, and adverse-risk groups. We chose to use data from the RNA-seq assay rather than from standard clinical testing, because our data above indicate that the RNA-seq assay was more accurate in identifying fusions and relevant pathogenic variants, and thus would provide for more robust stratification. For this stratification, we considered outlier *MECOM* expression to be an adverse prognostic marker, in lieu of detecting inv(3) rearrangements. We then divided the patient cohorts into terciles based on APS value, noting that the APS values for favorable-, intermediate-, and adverse-risk patients showed a significant increase with increasing

risk status (Fig. 4D). We then used these APS tercile values to re-stratify the patients as follows: patients with first-tercile (low-risk) APS values were stratified as favorable-risk, patients with second-tercile APS values retained their ELN-RNA stratification, and patients with third-tercile (high-risk) APS values were stratified as adverse-risk.

In the AML PMP cohort, 34 of 154 (22.1%) patients were re-stratified (Fig. 5A). As expected, the number of intermediate-risk patients in the ELN-RNA-APS model was reduced—from 38 in the ELN-RNA model to 13 in the ELN-RNA-APS model. Six patients moved from the favorable-risk group to adverse—a tAML case with inv(16) cytogenetics, and five other cases with *NPM1* mutations. The four cases that moved from the intermediate to favorable categories all had long survival durations, and three of the four carried both *NPM1* and *FLT3*-ITD mutations. Twenty-one cases were moved from the intermediate to adverse risk groups. This set of patients was characterized by tAML and sAML disease, and several normal-karyotype AML patients with no detected fusions or alterations (all with very poor outcomes). Finally, three patients transitioned from the adverse to favorable category—these patients had either *KMT2A*-PTD alterations or a *NUP98-KMT2A* gene fusion. Notably, the survival curves for the ELN-RNA-APS model were not different from the ELN-RNA model, indicating that the patients re-stratified away from intermediate status were accurately re-assigned (Fig. 5B–C).

We then applied the same models to the TCGA LAML and Beat AML cohorts (Fig. 5D–I). In both validation cohorts, we observed a similar pattern: the number of intermediate-risk

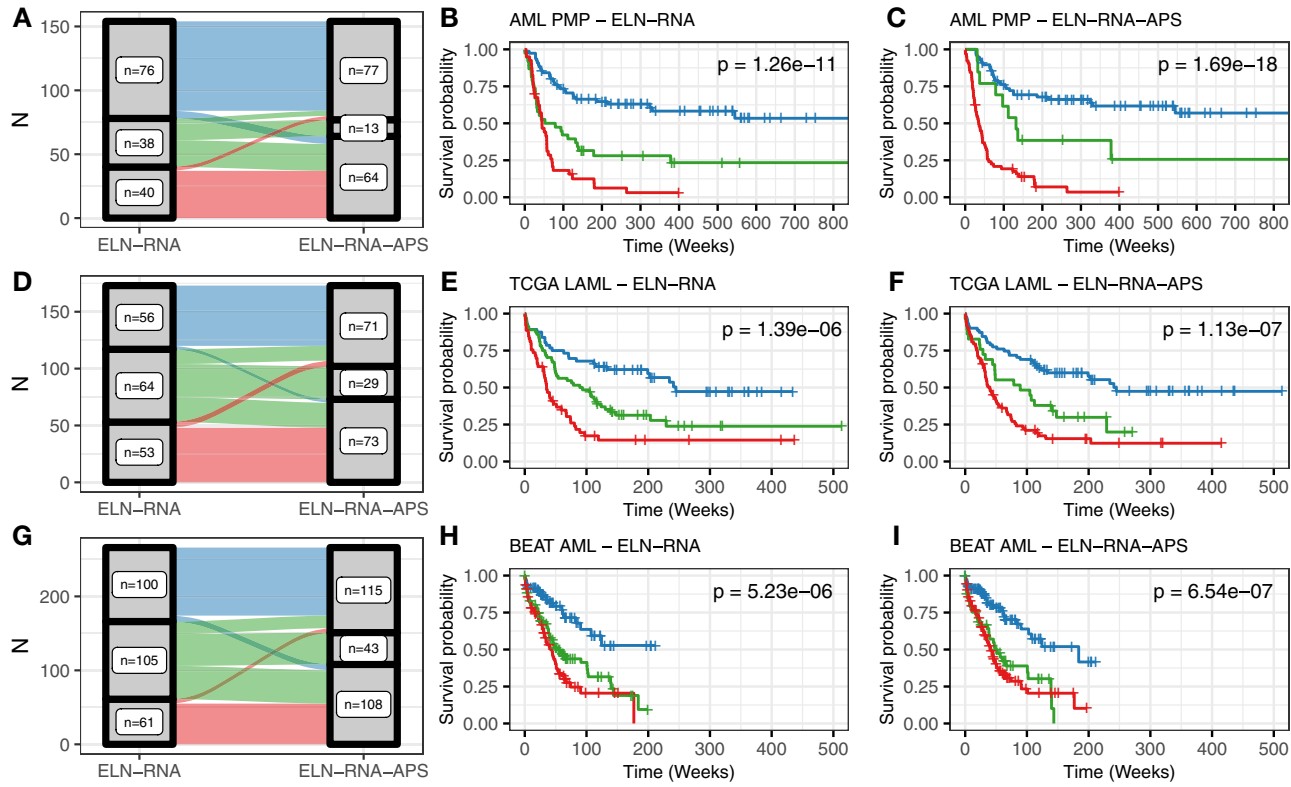

**Fig. 5 Patient stratification. A** Comparison of patient stratification by ELN-RNA and ELN-RNA-APS models for the AML PMP cohort. The width of each link is scaled to the number of patients shared between each set of risk categories. **B–C** Survival curves for the ELN-RNA and ELN-RNA-APS models for the AML PMP cohort. **D–I** Stratification and survival curves for the TCGA LAML and Beat AML patient cohorts. Patient groups are colored as in Fig. 4D (favorable—blue, intermediate—green, adverse—red), with log-rank p values indicated.

patients was reduced, while the outcomes for the newly assigned adverse-risk and favorable-risk patients were quite similar to the ELN-RNA model. In the TCGA LAML cohort (Fig. 5D), 43 of 173 (24.9%) cases were reassigned, while in the Beat AML cohort (Fig. 5G), 74 of 293 (25.3%) were reassigned. Re-stratified cases were similar in nature to those described in the AML PMP cohort, with similar survival results. Altogether, in all three cohorts, the addition of expression information in the ELN-RNA-APS stratification model provided a clear stratification improvement over the standard model using ELN criteria alone.

To determine the impact of including clinical karyotyping information instead of the purely RNA-Seq based stratification models, we generated alternate patient stratification models which used the diagnostic karyotype information rather than gene fusion data as the source of SV information (Supplementary Fig. 15). While 22/154 cases were discrepant between the stratification models relying on molecular alterations alone, applying the APS re-stratification reduced the number of discrepancies to 9/154. Of these nine cases, two were discrepant due to cryptic *KMT2A*-family fusions observed in the RNA-Seq data (and so were identified as adverse by the RNA-Seq based models)—these patients both showed very poor outcomes (mean survival time of 22 weeks). The remaining seven were discrepant due to rare cytogenetic abnormalities without corresponding gene fusion events (and so were identified as adverse by the karyotype-based models)—these patients showed better outcomes (median survival time of 111 weeks). Though the number of cases is low, these results suggest that including diagnostic karyotyping information would not necessarily improve the RNA-Seq based stratification model.

**Pathway analysis and differential expression analysis**. To demonstrate the utility of the transcriptome-based assay and gene expression signature for therapy selection, we first used Ingenuity Pathway Analysis (IPA)[41] and Gene Set Enrichment Analysis (GSEA)[42] to identify differentially activated pathways between patients with first-tercile (low-risk) and third-tercile (high-risk) APS values, excluding acute promyelocytic leukemia (APL) patients. In the IPA analysis, we observed results consistent with dysregulation of molecules involved in integrin, chemokine, and cytokine signaling in the high-risk group (Fig. 6A). We performed a similar analysis using GSEA (with the Reactome[43] database of pathways) and saw similar results: integrin and chemokine signaling pathways were strongly enriched across all three patient cohorts in the high-risk group (Fig. 6B). To identify potentially targetable molecules from the pathway enrichment results, we analyzed molecules that recurred across multiple enriched pathways or gene sets. This analysis revealed integrins (e.g., *ITGB3*, *ITGA2B*), focal adhesions and cytoskeletal rearrangements (e.g., *PTK2*, *MYLK*, *MYL9*), SRC/RAS signaling components (e.g., *SRC*, *RASD1*, *RASGRP1*), chemokines (e.g., *CXCL1*, *CCR4*, *CXCL3*), and PI3K/AKT/MTOR signaling (e.g., *AKT3*) as being some of the key dysregulated molecules in third-tercile APS patients (Supplementary Fig. 16).

To further analyze the biological mechanisms underlying the differences between patients with low and high APS values, we performed differential expression analysis (Fig. 7). We observed large differences for the APS genes *CD109*, *CALCRL*, and *TMEM273*. Interestingly, *CD109* was previously identified as being upregulated in *RUNX1*-mutated AMLs[44], and *CALCRL* has also recently been described as being over-expressed in the immature CD34+CD38− compartment of AML patient marrow[45]. Previously

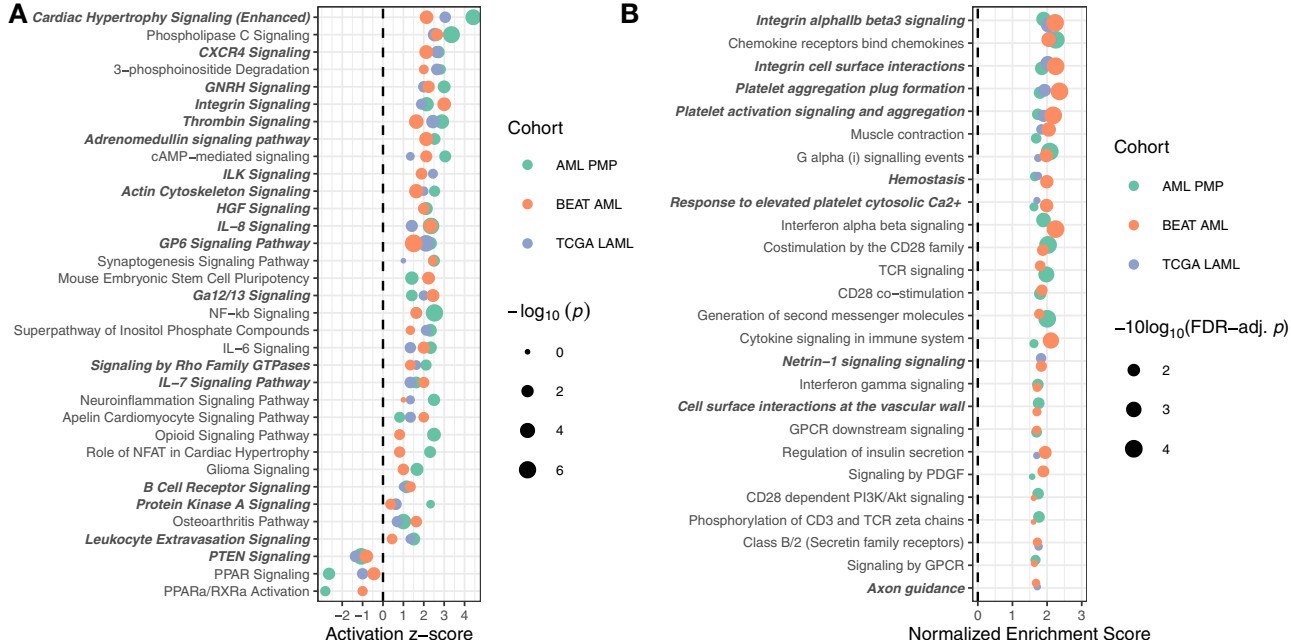

**Fig. 6 Pathway enrichment analysis. A** Pathway enrichment analysis of IPA canonical pathways overrepresented in differentially expressed genes across all three cohorts. Pathways are ranked by mean activation z-score, with point size scaled by Fisher's exact test p values. **B** Geneset enrichment analysis (GSEA) against the Reactome pathway database. Pathways are ranked by the number of cohorts the pathway was enriched in, then by the mean FDR-adjusted p value across cohorts. GSEA p values are derived from permutation testing, and corrected for multiple testing using the FDR method. In both panels, gene sets containing *PTK2* and/or *ITGB3* are highlighted in bold and italic.

described genes with prognostic implications in AML such as *GPR56*[38] and *MECOM*[8,31] were also consistently upregulated in the third-tercile set of patients. Among the most downregulated genes, we observed *TRH* (previously described as being associated with t(8;21) AML)[46], *IL5RA*, and the heme peroxidases *LPO* and *MPO*.

As a proof-of-principle for using transcriptome-based testing for therapy selection, we chose to focus on integrin signaling via FAK. In particular, we observed that *ITGB3* and *PTK2* (which encodes FAK), were highly expressed in third-tercile APS patients in all three cohorts, and were central molecules in many of the enriched pathways. FAK has been previously characterized as a therapeutic target in various cancers[47–49], while *ITGB3* has been demonstrated to be a vulnerability in some murine leukemias[50].

**Dysregulated integrin signaling in high-risk AML.** Based on the pathway and differential expression analysis, we sought to determine whether the high-risk AML cases with dysregulated integrin signaling were characterized by specific genetic lesions. In all three cohorts, we observed a significant concentration of *RUNX1* or *TP53* mutation in patients with elevated expression of *PTK2* (Fig. 8A). In addition, elevated expression of *PTK2* often occurred in patients with high APS values, and with sAML or tAML (Supplementary Fig. 17). Interestingly, high expression of *PTK2* was not associated with *FLT3*-ITD mutation, indicating that this overexpression was associated with a subset of high-risk AML cases, rather than all high-risk AML. We also observed, using data from the AML Proteome Atlas[51], that FAK protein expression was strongly correlated with the expression of several other proteins involved in focal adhesions, including SRC, ITGB3, and ITGA2 (Fig. 8B). In addition, protein expression of FAK was significantly higher in patients with *RUNX1* mutation, and also elevated (non-significantly) in patients with *TP53* mutation (Fig. 8C), but decreased in patients with *FLT3* mutations, confirming that the observations in the RNA-Seq data were also present at the protein level.

We then sought to test whether inhibition of FAK in cell line model systems could be an effective therapeutic strategy for *RUNX1*- or *TP53*-mutated cells. First, in the MDSL cell line model[52,53], we observed that CRISPR-mediated inactivation of *RUNX1* or *TP53* led to elevated FAK expression and increased sensitivity to the FAK inhibitor defactinib (Supplementary Fig. 18). Next, we tested the AML cell lines KG1a and THP-1, following shRNA-mediated *RUNX1* knockdown. Depletion of *RUNX1* reduced the levels of RUNX1 protein and induced levels of FAK (Fig. 8D). *RUNX1* knockdown sensitized cells to inhibition by the FAK inhibitors VS-4718 and defactinib (Fig. 8E–F), suggesting that these inhibitors may be efficacious in AML cases with loss-of-function *RUNX1* and possibly *TP53* mutations.

Together, these results demonstrate a rationale for additional testing of FAK inhibitors in AML patients with *RUNX1* or *TP53* mutation. Further, our analyses revealed that some APS-high cases that did not exhibit a *RUNX1* or *TP53* variant also showed increased *PTK2* expression, suggesting that transcriptome-based testing, in addition to better risk stratification, could permit better selection of therapies for myeloid malignancies. In the meantime, our approach reveals potential genetic variant biomarkers that could assist in selecting AML patients for FAK inhibitor clinical trials in AML.

**Discussion**

We compared RNA-Seq to WES and WGS-based approaches and found RNA-Seq to be superior for clinical assessment of myeloid malignancies. The RNA-Seq assay was capable of recovering (either directly or indirectly) all the clinically relevant chromosomal translocations and inversions in the exploratory patient cohort, as well as identifying novel translocations. The RNA-Seq assay showed improved accuracy for recovering *FLT3*-ITD events at low VAF. We also observed that, rather than suffering compared to WES for detection of clinically relevant SNV and short

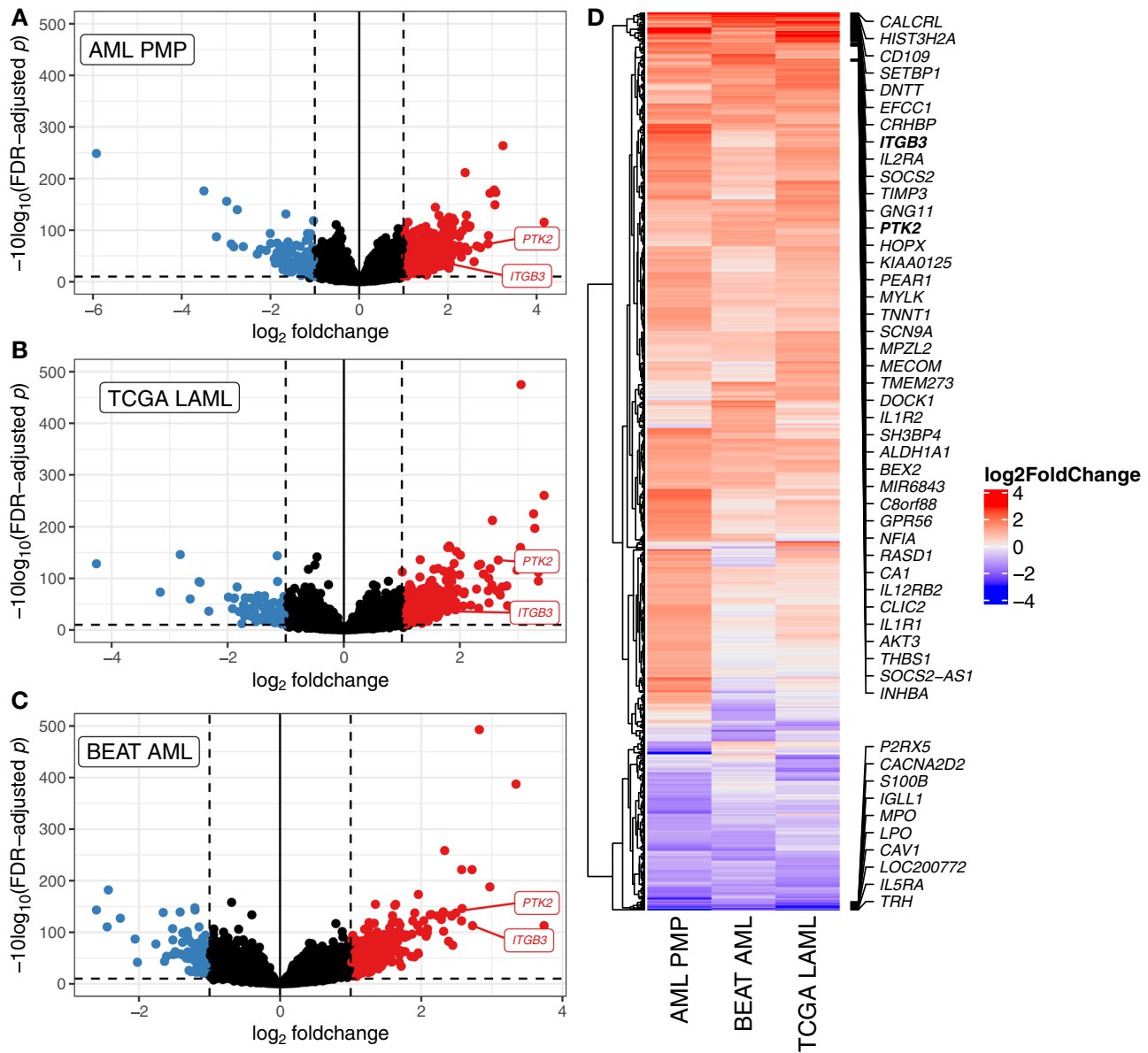

**Fig. 7 Differential gene expression for patients with first-tercile vs. third-tercile APS values. A–C** Volcano plots of differentially expressed genes for the AML PMP (**A**), TCGA LAML (**B**), and Beat AML (**C**) cohorts. Genes with absolute $\log_2$fold-change of 1 and FDR-adjusted $p \leq 0.1$ (based on Wald tests as implemented in DESeq2) are highlighted in red (over-expressed in third-tercile APS) and blue (over-expressed in first-tercile APS). **D** Comparison of differentially expressed genes between cohorts. All genes which were differentially expressed in any single cohort are displayed with their log2FoldChange values across all cohorts. The top 40 over-expressed and bottom ten under-expressed genes (by mean fold-change across cohorts) are labeled.

indel variants, the RNA-Seq assay had higher sensitivity (for coding regions in myeloid-relevant targets). Our validation analyses demonstrated that the reproducibility and analytic validity of RNA-Seq meet the necessary standards for clinical implementation. While not all variants are expressed in bulk RNA-Seq data[54], clinically relevant driver mutations are invariably present. Due to the high sensitivity of the assay for detection of alterations in *FLT3* and *IDH1/IDH2*, the RNA-Seq assay could be used to accurately assign patients to therapies targeting those genes. Indeed, since the RNA-Seq assay provides a wealth of additional information about the size, breakpoints, and expression level of *FLT3*-ITD events (compared to PCR- or panel-based assays), the assay could be used to prospectively determine the relationship between these parameters and response to *FLT3* inhibition.

Through expression-signature-based approaches, we saw that incorporation of gene expression information offers substantial

benefits in terms of identifying lower- and higher-risk patients, and the potential for identifying patients who may respond to specific therapies. In particular, the APS value is a hazard indicator of similar magnitude to *TP53* mutation, with high values indicating a very poor prognosis. Intriguingly, the APS signature was also strongly prognostic for overall survival when applied to pediatric AML cases, suggesting that even though the mutational spectra are somewhat different between adult and pediatric AML, similar biological pathways drive AML in children and adults. The APS signature outperformed the LSC17 score with respect to predicting outcome, and there are at least three potential explanations for this. First, the improved performance of the APS signature may be due to the use of the entirety of the transcriptome to generate the model, thus allowing for the capture of cell nonautonomous signals from the microenvironment, as opposed to the LSC17 score, which is based solely on a leukemic

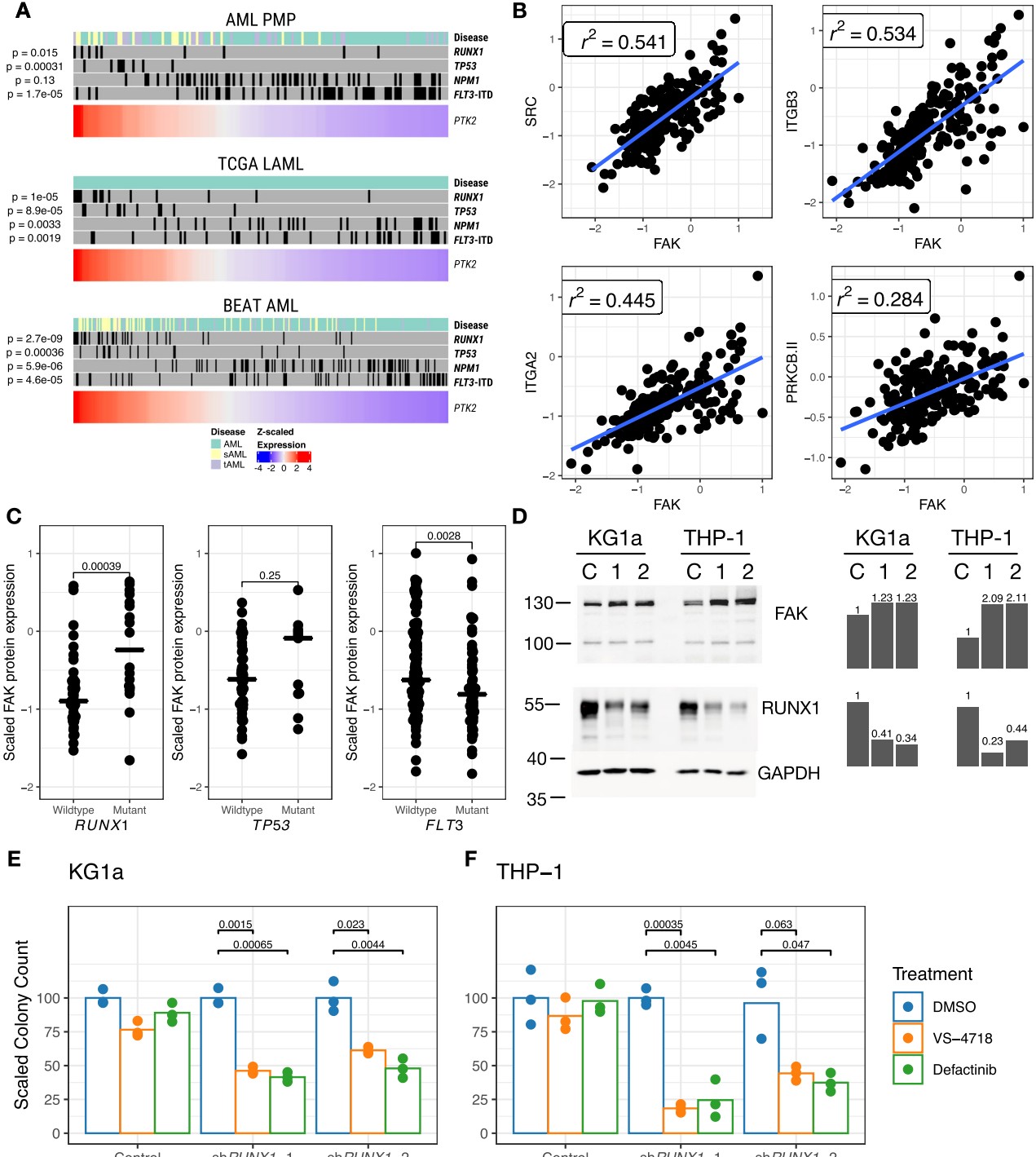

**Fig. 8 Correlation of *PTK2* expression with specific mutations. A** Samples from each cohort ranked by z-scaled expression of *PTK2*. The upper annotation plot for each cohort indicates the presence of selected mutations. Indicated *p* values are for Kruskal-Wallis tests comparing continuous *PTK2* expression against presence or absence of specific mutations. **B** Top highly correlated proteins with FAK. $r^2$ values indicate Pearson correlation coefficients. **C** Relative FAK protein expression in patients with and without mutant *RUNX1*, *TP53*, or *FLT3* in the AML Proteome Atlas. Two-sided *t*-test *p*-values are indicated for each comparison. **D** Western blot assessment of FAK and RUNX1 expression in KG1a and THP-1 cell line derivatives. Cell lines were treated with either (**C**) control shRNA, (1) sh*RUNX1*-1, or (2) sh*RUNX1*-2, with molecular weights quantified in kilodaltons. Quantifications scaled to controls are shown as bar plots. Each blot represents a single experiment, with the exception of RUNX1 in THP-1, which was performed twice with similar results. **E**, **F** Colony-forming cell assay for AML cell lines with short-hairpin RNAs against *RUNX1*, and treated with FAK inhibitors VS-4718 (0.5 μm for KG1a, 1.5 μm for THP-1) or Defactinib (1 μm for both cell lines). The indicated *p* values correspond to two-sided *t*-tests for each comparison.

stem cell signature. Second, recent studies have demonstrated the potential for mature leukemia cells to de-differentiate and re-gain stem-like properties[55]. Since the APS signature was trained on bulk peripheral blood or bone marrow patient material, it could capture signals identifying those cases with nascent de-differentiation potential that are likely to progress. Third, since the APS signature was trained using RNA-Seq data rather than microarray data, it might benefit from the improved resolution of RNA-Seq. Our approach highlights the benefit of incorporating gene expression analysis into the clinical assessment of myeloid malignancies, and the utility of integrative analysis of genomic data for stratification and therapy selection. In the near term, this clinically validated RNA-Seq assay would significantly improve the stratification of patients to stem-cell transplant.

Our work also serves as a proof-of-concept for the use of a clinical RNA-Seq assay to identify activated or inactivated pathways, potentially enabling predictions about the susceptibility of AML patients to specific targeted therapies. In particular, we show the correlation of integrin pathway activation with *RUNX1* or *TP53* variants, but not with other high-risk AMLs expressing *FLT3*-ITD variants. Given the extremely poor outcome of AML with either of these mutations, and the lack of current therapies directed against these variants, our data suggest a potential role for FAK inhibitors in treating AML with *RUNX1* and possibly *TP53* variants. However, we also show that *PTK2* and integrin activation occurs in cases lacking *RUNX1* or *TP53* variants, which arguably makes RNA-Seq the clinical assay of choice in high-risk AML. We suggest that a clinical RNA-Seq assay could also be adapted for routine clinical use to identify patient-specific therapeutic options at diagnosis using the previously analyzed cohorts as comparators.

## Methods

**Study design**. We generated sequence data from an exploratory cohort of 176 patients, generating RNA-Seq libraries for all patients (Fig. 1A, Supplementary Fig. 1). The exploratory patient cohort spanned a range of myeloid malignancies, including AML, MDS, sAML, tAML, and tMDS (Supplementary Data 1–4). We sequenced material from patient bone marrow, peripheral blood, or leukapheresis samples collected at diagnosis. For most samples, prior clinical test results were available (though there were many cases of missing data, and not all patients received the same tests as the standard of care evolved through time). The exploratory RNA-Seq libraries were prepared in two separate batches, referred to as 'batch one' and batch two'. To compare RNA-Seq against alternate technological platforms, we generated WGS data for 18 samples and WES data for 89 samples (Supplementary Data 2).

We then constructed an informatic pipeline to detect and annotate SVs, SNVs, short indels, and gene expression counts for all samples (as appropriate for each platform, Supplementary Fig. 1B). SVs were assessed in an unbiased fashion, while SNVs and short indels were inferred for a set of curated genes with known clinical relevance in myeloid malignancies (Supplementary Data 9). For the SV and SNV/short indel analyses, we used the entire exploratory batch of 176 cases, while with subsequent gene expression and stratification analyses we used only the 154 AML, sAML, and tAML cases, as these analyses were intended to be AML-specific (Fig. 1A). To determine the reproducibility of the analysis, we prepared and sequenced material from a validation cohort consisting of repeated patient samples and commercially obtained cell line material, prepared as technical replicates in triplicate from single RNA extractions (Supplementary Data 5, 14).

We also re-analyzed samples from the TCGA LAML[19] and Beat AML[20] cohorts (Supplementary Data 7-8). Sample details for the TCGA LAML cohort were retrieved from the National Cancer Institute Genomic Data Commons (GDC) [https://portal.gdc.cancer.gov/projects/TCGA-LAML], cBioportal [https://www.cbioportal.org/study/summary?id=laml_tcga_pub], the Broad GDAC Firehose [https://gdac.broadinstitute.org], and from the Tumor Fusion Gene Data Portal [https://tumorfusions.org][56]. For data from the Beat AML project, see the Data Availability section of Tyner et al[20]. For both the TCGA LAML and Beat AML cohorts, we used the published SNV, short indel, and SV calls, but performed gene expression analyses using the pipeline described below. To further evaluate the derived gene expression signature, we analyzed data from an additional local prospective cohort of 42 cases (Supplementary Data 6). In order to determine whether the APS was applicable to pediatric AML, we analyzed data from the TARGET pediatric AML project[21], which was also retrieved from the GDC [https://portal.gdc.cancer.gov/projects/TARGET-AML]. Matched ribo-depleted libraries for four cases were prepared by the Centre for Epigenome Mapping

platform, available via European Genome-Phenome Archive Study EGAS00001000552 [https://www.ebi.ac.uk/ega/studies/EGAS00001000552].

**Ethics approval and consent to participate**. Peripheral blood, bone marrow, and leukapheresis samples were obtained from consenting patients via the Hematology Cell Bank of British Columbia [http://hematology.med.ubc.ca/research/hematology-cell-bank-of-bc/]. Ethics protocols were approved by the BC Cancer Research Ethics Board, under protocols H04-61292, H09-01779, H11-01484, and H13-02687.

**RNA extraction, library construction, and sequencing**. RNA was manually extracted from bone marrow or peripheral blood using Qiagen Allprep kits. Total RNA samples were checked using Agilent Bioanalyzer RNA nanochip or Caliper GX HT RNA LabChip. Samples that passed quality check were arrayed into a 96-well plate. Following this, polyA+ RNA was purified using the 96-well MultiMACS mRNA isolation kit on the MultiMACS 96 separator (Miltenyi Biotec, Germany) from total RNA with on column DNaseI-treatment as per the manufacturer's instructions. The eluted polyA+ RNA was ethanol precipitated and resuspended in 10 μL of DEPC-treated water with 1:20 SuperaseIN (Life Technologies, USA).

Double-stranded cDNA was synthesized from the purified polyA+ RNA using the Maxima H Minus First Strand cDNA Synthesis Kit (Thermo Fisher Scientific Inc., USA) and random hexamer primers. Quality passed cDNA plate was fragmented by Covaris LE220 for 2 × 65 s at duty cycle of 30%. The paired-end sequencing library was prepared following Canada's Michael Smith Genome Sciences Centre paired-end library preparation plate-based library construction protocol on a Biomek FX robot (Beckman-Coulter, USA). Briefly, the cDNA was subject to end-repair, and phosphorylation by T4 DNA polymerase, Klenow DNA Polymerase, and T4 polynucleotide kinase respectively in a single reaction, followed by cleanup using magnetic beads and 3′ A-tailing by Klenow fragment (3′ to 5′ exo minus). After cleanup, adapter ligation was performed. The adapter-ligated products were purified using magnetic beads, then UNG digested and PCR-amplified with Phusion DNA Polymerase (Thermo Fisher Scientific Inc., USA) using Illumina's PE primer set in a single reaction, with cycle condition 37 °C 15 min, 98 °C 1 min followed by 13 cycles of 98 °C 15 s, 65 °C 30 s and 72 °C 30 s, and then 72 °C 5 min. The PCR products were purified and size selected using magnetic beads, checked with Caliper LabChip GX for DNA samples using the High Sensitivity Assay (PerkinElmer, Inc. USA) and quantified with the Quant-iT dsDNA HS Assay Kit using Qubit fluorometer (Invitrogen). Libraries were normalized and pooled. The final concentration was determined by Qubit dsDNA HS Assay for Illumina Sequencing.

For the first exploratory cohort RNA-Seq batch, we sequenced a single library per Illumina HiSeq 2000 lane, using 2 × 75 bp reads, which resulted in ~400 million reads per library. Ninety-two samples were initially submitted for library construction, of which 89 were successfully prepared and sequenced. For the second exploratory cohort RNA-Seq batch, we sequenced two libraries per Illumina HiSeq 2000 lane, using 2 × 75 bp reads, which resulted in ~200 million reads per library. Ninety-two samples were initially submitted for library construction, of which 87 were successfully prepared and sequenced. For the validation and prospective RNA-Seq cohorts, we sequenced two libraries per Illumina HiSeq 2000 lane, using 2 × 100 bp reads, which resulted in ~200 million reads per library. For a set of three samples, we sequenced two HiSeq lanes. Seventy-one samples were successfully prepared and sequenced for the validation cohort, and 42 for the prospective cohort. For both the validation and prospective cohorts, a modified strand-specific RNA-Seq library construction protocol was used.

**DNA extraction, library construction, and sequencing**. DNA was extracted from bone marrow or peripheral blood using Qiagen Allprep kits. Genome libraries with fragment size ranges of ~400 bp were constructed on a SPRI-TE robot (Beckman Coulter, USA) according to the manufacturer's instructions (SPRIworks Fragment Library System I Kit, A84801). Briefly, 1 μg of genomic DNA in a 60 μL volume, and 96-well format, was fragmented by Covaris E210 sonication for 30 s using a duty cycle of 20% and intensity of 5. Up to 10 paired-end genome sequencing libraries were prepared in parallel using the SPRI-TE 300-600 bp size-selection program. Following completion of the SPRI-TE run the adapter-ligated library templates were quantified using a Qubit fluorometer. Five nanograms of adapter-ligated template was PCR amplified using Phusion DNA Polymerase (Thermo Fisher Scientific Inc. USA) and Illumina's PE indexed primer set, with cycle conditions: 98 °C for 30 s followed by 10 cycles of 98 °C for 15 s, 62 °C for 30 s and 72 °C for 30 s, and a final amplicon extension at 72 °C for 5 min. The PCR products were purified using Ampure XP SPRI beads, and analyzed with Caliper LabChip GX for DNA samples using the High Sensitivity Assay (PerkinElmer, Inc. USA). PCR products of the desired size range were purified using gel electrophoresis (8% PAGE or 1.5% Metaphor agarose gels in a custom built robot) and the DNA quality was assessed and quantified using an Agilent DNA 1000 series II assay and Quant-iT dsDNA HS Assay Kit using Qubit fluorometer (Invitrogen), then diluted to 8 nM. The final concentration was verified by Quant-iT dsDNA HS Assay prior to Illumina Sequencing.

Two hundred and fifty nanograms of the constructed genome libraries were used to capture the exome using Agilent SureSelect Human All Exon (50 Mb)

capture probes. 89 exome libraries were sequenced with paired-end 100 bp reads (1 library per HiSeq 2000 lane). 20 WGS libraries were sequenced with paired-end 100 bp reads on the Illumina HiSeq 2000 instrument, sequencing three lanes per sample.

**Bioinformatics pipeline overview**. All samples were processed using customized in-house bioinformatics pipelines, as well as the bcbio-nextgen pipeline framework [http://bcbio-nextgen.readthedocs.io/]. The WGS and WES data was initially processed by the Bioinformatics Core at Canada's Michael Smith Genome Sciences Centre, Vancouver, Canada, while the RNA-Seq data was processed by the authors. All variant-calling and tertiary analyses were performed by the authors.

**Read alignment and quality control**. WGS and WES sequence data were aligned to the hg19 human reference genome using BWA (version 0.5.7, 'aln' and 'sampe' subcommands)[57], with duplicates marked using Picard [http://broadinstitute.github.io/picard/]. RNA-Seq data was aligned using GSNAP (version 2013-10-28)[58], using the hg19 reference and with command-line arguments '–novelsplicing 1 –max-mismatches 10 –use-splicing'.

We used RNA-SeQC (version v1.1.8)[59] to gather quality metrics for the RNA-Seq libraries. To examine the range of variation and identify outliers in our data set, we constructed Levey-Jennings charts[60] for all the metrics provided by RNA-SeQC (Supplementary Fig. 2). These charts identified several samples which were consistent outliers (metric values exceeding plus or minus two standard deviations from the mean). From the available RNA-SeQC metrics, we selected six key variables (number of mapped reads, mapping rate, rRNA rate, intragenic rate, mean fragment length, and exonic rate). We then failed all libraries which had outlying values (plus or minus two standard deviations from the mean) for three or more of those variables. The failed validation cohort libraries all corresponded to cases where we attempted to re-sequence patient material, starting from limited amounts of input RNA. Of the original 176 libraries in the exploratory cohort, three samples were failed, while six samples from the validation cohort were failed due to poor library quality. The failures were generally associated with poor input material quality as most of the failed libraries came from attempts to re-generate additional sequencing libraries from older material.

**SNV and small indel detection**. We used the bcbio-nextgen pipeline (version 1.0.1) to call variants using three callers: VarScan (version 2.4.2)[24], GATK HaplotypeCaller (version 3.7)[22], and FreeBayes (version 1.1.0)[23]. In all cases, the variant-calling region was restricted to the clinical targets of interest (Supplementary Data 9). For these targets, we either selected all coding exons, or specific 'hot-spot' targets to assess for SNVs and short indels. All variant callers were configured as specified by the bcbio pipeline software. In addition to the three variant callers, we used the bcbio ensemble tool, configured to report a consensus variant file containing all variants called by at least two callers, without being filtered by any single caller. We used vt (version 0.57)[61] to decompose multiallelic calls, and to normalize indel positions. All variants were then annotated using snpEff (version 4.3 g)[62], with HGVS nomenclature generated using a single chosen transcript model for each gene. We used gemini (version 0.20.0)[63] to further annotate variants against reference data sets including COSMIC (version 68))[64], ExAC (r0.3) and gnomAD (r2.0.1)[65], ClinVar (v20160502)[66], and CADD (version 1.0)[67].

We first determined the clinically relevant SNVs and short indel alterations for each patient. Briefly, from the unfiltered variant calls, we filtered coding variants with moderate ExAC[65] population frequencies (≥0.01 for synonymous variants, ≥0.1 for missense variants), and recurrent low-quality and artefactual variants. We validated the presence of selected mutations (for 157 cases where the material was available) using mononuclear cells (153 cases) or red cell lysed marrow cells (four cases), and determined the somatic status for variants where patient material was available (147 cases) using cultured marrow fibroblasts (107 cases) or sorted CD3$^+$ T cells (40 cases) from the same patient specimen, using targeted sequencing experiments. This validation was carried out by Canada's Michael Smith Genome Sciences Centre, with PCR primer design and data reviewed by the authors.

Since the raw call set for the RNA-Seq libraries contained many splice-alignment artefacts, we filtered all variants annotated as intronic or in splice donor/acceptor sites. All calls from the ensemble set were retained, as well as selected calls reported by only a single caller, as determined during the report review. Additional recurrent artefactual variants in the RNA-Seq data were manually reviewed and removed if they were determined to be both recurrent and the result of splice-alignment artefacts, polyA/T runs, or poor base quality. For some cases where complex indel calls were reported with slight nomenclature differences between callers, coordinates were manually reviewed and merged if they reported identical events (e.g., for an indel in CEBPA with calls from GATK-haplotype (c.283_288delGTGGGC and c.292delA) and FreeBayes (c.283_289delGTGGGCC and c.292A > C), the nomenclature was adjusted to allow the two calls to be counted as concordant with each other). We additionally labeled all variants that were either (a) synonymous with adjusted ExAC population frequencies ≥0.01 or (b) missense variants with adjusted ExAC population frequencies ≥0.1 as benign.

To compare variant call sets between matched RNA-Seq and WES or WGS libraries, we generated merged variant sets for each relevant library pair, using the

ensemble call set. We used bedtools (version 2.24.0)[68] and VarScan2 (v2.4.2)[24] to generate coverage depth and the observed non-reference allele frequency across all SNV targets (Supplementary Data 9). We used a coverage threshold of 50× for reporting low coverage, and 100× for a warning threshold in the RNA-Seq data. In addition, we used the single-caller calls for each caller and library to compare differences between callers. For these comparisons, we considered sites with <10-fold coverage to be effectively uncallable. For several sites in the RNA-Seq data where the matched WES/WGS data indicated a potential missed call, we identified an alignment issue in the pipeline that was addressed in a software update and so did not consider these sites as discordant.

To analyze libraries from the validation cohort, we retained all calls from all callers (i.e., keeping common synonymous variants), with the exceptions of known recurrent artefacts, SNV calls near FLT3-ITD boundaries, and sites with coverage <10-fold coverage (Supplementary Data 15). Note that for samples 157-18 and 213-51, two of three replicates failed sequencing quality checks, therefore these samples were excluded from further analysis. In the raw call set, we observed several false-negative and false-positive calls that were due to low sequence coverage of the gene of interest (which would be reported as uncallable by our standard procedure) or one of eight recurrent spurious artefacts (which were added to a list of known artefacts for the pipeline). For example, CEBPA expression is very low in the NIST standard cell line NA12878[69] (Supplementary Fig. 6) compared to patient material. We thus filtered these calls from the dataset (Supplementary Data 15), and examined the remaining true-positive, false-positive, and false-negative calls (Supplementary Fig. 7). We calculated the number of true positive, false positive, and false negative sites as the average among replicates for each cell line or repeated patient sample. In addition, we calculated the coefficient of variation between replicates in terms of the number of retained variant calls per replicate.

**Structural variation**. To detect inter- and intra-gene SVs, we performed de novo assembly on all the RNA-Seq libraries, using Trans-ABySS (version 1.5.2)[70] and PAVfinder (versions 0.2.0 and 0.3.0)[71]. PAVfinder relied on GMAP (version 2014-12-28)[58] and BWA MEM (version 0.7.12-r1039)[72] for contig-to-genome, contig-to-transcript, and read-to-contig alignments. The parameters used for trans-ABySS included assembly at three values of $k$ (the kmer size used for assembly)—32, 52, and 72. These three assemblies were merged before further analysis with PAVfinder. We used custom R[73] and Python tools to annotate all events with gene expression estimates, to annotate calls with similar coordinates to each other, and allow for annotation of known recurrent artefactual and common events.

We then filtered the raw event lists as follows. 'Recurrent Artefact' and 'Recurrent Common' events were identified based on manual review of recurrence within the cohort, fusion properties, and literature review (Supplementary Data 20), and were removed. We then collapsed similar events (i.e., events with the same gene partners) into single events, and filtered out events matching any of these criteria: minimal support (fewer than eight supporting reads), average TPM expression value for the two fusion partners <2 or >130), inter-gene distance for putative fusion partners on the same chromosome of $<2 \times 10^6$ bp, or putative fusions involving HBA2.

We then constructed a list of expected gene fusions present in the exploratory cohort using the cytogenetics nomenclature and determined whether those events were captured in the RNA-Seq data. To construct the set of novel gene fusions used for patient stratification and downstream analysis, the set of fusion results was further filtered by manual review, selecting gene fusions with strong supporting evidence and/or genes previously known to be involved in myeloid malignancies (Supplementary Data 21–22).

To further verify the presence of the reported gene fusions, we performed the following additional validation steps. All of the retained fusions in disease-related genes (Supplementary Data 21) were manually reviewed for supporting read evidence in IGV. In addition, where copy-number array, WES, WGS, or gene panel data was available, those data sources were also reviewed. Due to limited patient material availability, validation of the remaining cases was not able to be performed. This review identified two sets of fusion partners (t(5;12) IL31RA-CTDSP2; t(X;Y) CD99P1-CD99) that were likely to be false-positive results—these results were removed from downstream analysis.

The reference truth set for FLT3-ITD calls was constructed principally from prior research testing results using a PCR-based assay. The complete set of PAVfinder results for each RNA-Seq library was filtered for events annotated as 'ITD' or 'ins' (for insertion) within FLT3. Each event was then annotated with the expression of FLT3 in transcripts per million (TPM). For each library in the exploratory cohort where a reference call was available, we assessed whether the PAVfinder output agreed or disagreed with the reference call. To estimate the relationship between RNA-Seq evidence and the allelic fraction of detected FLT3-ITD events, we performed linear regression analysis, modeling the observed VAF from GATK HaplotypeCaller (for those ITD events that were captured by that tool in the RNA-Seq and WES datasets) against combinations of spanning read support, ITD size, median coverage over the FLT3 ITD region, and expression of FLT3. The best model accuracy was obtained using only RNA-Seq VAF estimates, using a linear model predicting FLT3-ITD VAF = (spanning reads * 0.293) + (median FLT3 coverage * −0.0213) + 29.4. This regression was then used to estimate the VAF for all detected FLT3-ITD events. We then dichotomized FLT3-ITD events into 'FLT3-ITD-low' and 'FLT3-ITD-high' support, using a VAF cutoff of 0.33

(corresponding to a mutant:wildtype allelic ratio of 0.5). For selected *FLT3*-ITD events, we used panel-based sequencing to validate the events. For the remaining cases, the detected events were reviewed manually using IGV[74]. To detect *KMT2A*-PTD events, we filtered the complete event set for events annotated as 'ins', 'ITD', or 'PTD', with either 'gene1' or 'gene2' annotated as *KMT2A*. Two low-evidence ITD events were removed based on manual inspection. Again, we used panel-based sequencing to confirm that two additional low-evidence calls were truly false-positives in the initial RNA-Seq result set.

**Gene expression signatures.** Expression quantification was performed for all RNA-Seq libraries using sailfish (version 0.9.0)[75], using RefSeq gene models downloaded in Gene Transfer Format from the UCSC genome browser on 2014-08-21, with gene models from non-standard chromosome sequences removed. Both isoform- and gene-specific quantifications were generated, and both raw estimated counts, as well as transcripts-per-million (TPM) estimates were used in downstream analysis. For the TCGA LAML and Beat AML cohorts, we re-quantified gene expression using the same software and gene models.

From the raw, gene-level expression estimates, we filtered the data as follows (calculating thresholds separately for each cohort). First, genes with median expression ≤0 TPM were removed. Second, to normalize counts across samples, we converted counts from TPM to $\log_2(\text{TPM} + 1)$ and applied quantile normalization using the preprocessCore Bioconductor package (version 1.48.0)[76]. To visualize and determine outlier status for single genes, we calculated $z$-scaled expression estimates (where $z = (x - \mu)/\sigma$) for each gene by comparing the expression of that gene in a particular sample against the range of expression for that gene across a given patient cohort. For *MECOM* outlier analysis, we labeled all samples with $z$-scaled expression ≥3 as high outliers. For other genes with known prognostic relevance in AML (*FLT3*, *GPR56*, *BAALC*, *MN1*, and *BCAT1*), we used the $z$-scaled expression values as continuous variables for survival modeling. We calculated values for the LSC17 gene expression signature using the model coefficients described by Ng et al.[10], using the normalized, TPM-scaled expression estimates for the AML PMP and TCGA LAML cohorts. We then dichotomized each cohort, using the median LSC17 score within each cohort to define high and low scores.

We used the R glmnet package (version 4.0-2)[35,77] to perform LASSO regression to identify a subset of features that could be used to fit a Cox regression model. We first filtered the retained gene set by restricting the set to genes with a minimum expression of 0.5 (in $\log_2(\text{TPM} + 1)$ units) across the AML PMP, TCGA LAML, and Beat AML cohorts. We initially used 10-fold cross-validation, using the 154 AML-like samples from the exploratory cohort as the training set, using parameters "family = 'cox', maxit = 10000" to select the value for the $\lambda$ parameter with the minimum mean cross-validated error. We then used this $\lambda$ value to extract the final set of 16 genes and model coefficients (the APS gene expression signature, Supplementary Data 25). We then used the 'predict' function within the glmnet package to calculate model predictions (i.e., APS values) for each of the samples, dichotomizing patients by the median value into 'APS-High' and 'APS-Low' sets. APS values were applied to the TCGA LAML and Beat AML cohorts in a similar manner (and dichotomized using the median within-cohort APS values).

**Expression-based survival models.** To derive revised models for AML stratification that incorporated gene expression information, we first tested many potential predictors in univariate Cox proportional hazards analyses (Supplementary Data 27). To control for multiple testing, we adjusted the estimated $p$ values using the false-discovery rate (FDR) method[78]. To facilitate visualization, we transformed the adjusted $p$ values by calculating $-10\log_{10}(\text{adj. } p)$, using a threshold of 10 for significance (corresponding to an adjusted $p$ value of 0.10). Since none of the APL patients succumbed to the disease, we could not calculate a hazard ratio for the t(15;17) alteration in the AML PMP cohort.

**Patient stratification by standard and expression-based models.** Initial patient stratifications (ELN-RNA) were performed based ELN molecular guidelines[5], modified to incorporate equivalent information available through the RNA-Seq assay (Supplementary Data 28). In addition, we included t(15;17) translocations as a favorable risk marker, and PTD in *KMT2A* (*MLL*) as an adverse risk marker, based on their known strong influence on AML outcomes[6,34]. For the ELN-RNA-APS model, we re-stratified patients with first-tercile or third-tercile values for the APS value into the favorable and adverse risk categories, respectively. In each case, custom R functions were written to stratify patients based on the presence or absence of particular molecular markers. We then determined the numbers of patients with agreeing or disagreeing risk stratifications by the different criteria, and manually reviewed discordant cases. We estimated survival outcomes for each patient stratification model using the Kaplan–Meier estimator, using the R survival (version 3.2-7)[79] and survminer (version 0.4.8)[80] packages. To compare the RNA-Seq-based stratifications with stratifications utilizing diagnostic karyotyping information, we re-stratified the patients using karyotype information (rather than gene fusion information), for the ELN-Cyto model, and applied the same APS rules to generate the ELN-Cyto-APS model (Supplementary Fig. 15).

**Pathway and differential expression analysis.** We used IPA (IPA, December 2018 Release)[41] to perform a Core Analysis separately for each cohort, retaining all molecules meeting the significance and fold-change thresholds, with other analysis

parameters set to their default values. To compare enriched pathways across cohorts, we ranked the pathways by the number of cohorts they occurred in, then by the mean activation $z$-score (as calculated by IPA). We used GSEA (version 3.0)[42] to perform gene-set enrichment analysis for selected pathway datasets including the MSigDB hallmark gene set collection[81], the Kyoto Encyclopedia of Genes and Genomes (KEGG)[82], Reactome[43], the Gene Ontology resource[83], and Wikipathways[84]. For the GSEA analysis, we used sets of genes ranked by their fold-change values as input, gene sets from MSigDB v6.2, and command-line arguments '-norm meandiv -nperm 1000 -scoring_scheme weighted -set_max 500 -set_min 15'. We also performed exploratory analyses using EnrichR (version 2.1)[85]. To identify recurrently implicated molecules, we inspected the IPA enriched molecule lists and GSEA leading edge genes from the significantly enriched pathways, and identified molecules that occurred repeatedly across those pathways.

We performed differential expression analysis using DESeq2 (version 1.26.0)[86], comparing patients with first-tercile vs. third-tercile APS values from within each cohort (removing APL patients bearing t(15;17) translocations). For the DESeq2 analysis, we used the raw expression count data, with all parameters set to their default values. For considering genes to be significantly up- or downregulated for downstream analysis, we used thresholds of FDR-adjusted $p$ value of ≤0.1 and absolute $\log_2$ foldchange ≥1 for all downstream analyses.

**Proteomics data analysis.** Proteomics data were retrieved from the AML Proteome Atlas[51]. To identify proteins correlated with FAK protein expression, we calculated Pearson correlation coefficients for FAK and all other proteins in the dataset, and extracted the most highly correlated proteins. To compare FAK protein expression in samples with different mutational status, we extracted *RUNX1* and *TP53* mutation annotations from the experimental design, and plotted relative FAK expression between samples with and without those mutations, using two-sided $t$ tests to determine significance.

**Cell line experiments.** The *RUNX1*- and *TP53*-knockout MDSL cell lines were generated and confirmed in Martinez-Hoyer et al.[53]. For shRNA knockdown experiments, the shRNA lentiviral constructs targeting human *RUNX1* (shRUNX1#1 TRCN0000013660 and shRUNX1 #2 TRCN0000338427) were purchased from MilliporeSigma. The pLKO.1 backbones were modified to express EGFP in place of the puromycin resistance gene. Non-targeting shScramble in pLKO.1 vectors expressing EGFP were used as controls. The shRNA Lentiviruses were prepared in HEK293T/17 cells, as previously described[53] and titres were typically in the $10^9$ transducing units/ml range. KG1a and THP1 cells were resuspended in growth media at $5 \times 10^5$ cells/ml. Transduction was performed for 4–6 h in the presence of 8 µg/ml polybrene. After transduction, cells were washed and expanded in growth media for 72 h, EGFP+ cells were further purified by fluorescence-activated cell sorting.

For colony-forming cell assays (CFC), 500 lentivirus-transduced (GFP+) KG1a or THP1 cells were plated in methylcellulose containing h-IL3, h-erythropoietin, h-GM-CSF and h-SCF (H4434; StemCell technologies). CFC medium was mixed before plating with FAK inhibitor or DMSO as control. The colonies were scored after 7–10 days.

Western blotting was done as previously described[53]: cells were pelleted and lysed in RIPA buffer (25 mM Tris, 150 mM NaCl, 1% TX-100, 0.25% sodium deoxycholate, and 0.1% SDS), to which protease and phosphatase inhibitors were added. Cell extracts were centrifuged at $16,000 \times g$ at 4 °C, and the supernatant was boiled for 5 min with Laemmli buffer. Samples containing 20–50 µg protein were separated using SDS–PAGE and transferred at 100 V for 1 h to nitrocellulose membranes, which were subsequently blocked using 5% milk or BSA in Tris-buffered saline with Tween-20 (20 mM Tris, 137 mM NaCl, 0.1% Tween-20). Primary antibodies used were as follows: anti-RUNX1 (CST 4334, 1:1000), anti-FAK (CST 13009, 1:1000), anti-GAPDH (CST 2118, 1:5000).

For RT-qPCR, RNA was extracted using Qiagen AllPrep DNA/RNA mini kit according to the manufacturer's manual. For mRNA targets, cDNA was synthesized using Maxima reverse transcriptase and random hexamer (Thermo Scientific). RT-qPCR was performed using power SYBR green master mix (Thermo Fisher Scientific), with the results normalized to HPRT1 expression. Primers used for qPCR were as follows: PTK2_set 1 forward - GTCTGCCTTCGCTTCACG, reverse - GAATTTGTAACTGGAAGATGCAAG, PTK2_set 2 forward GCGTCT AATCCGACAGCAACA, reverse -CTCGAGAGAGTCTCACATCAGGTT, HPR T1 forward TGACCTTGATTTATTTTGCATACC, reverse CGAGCAAGACGT TCAGTCCT.

**Reporting summary.** Further information on research design is available in the Nature Research Reporting Summary linked to this article.

## Data availability
The sequencing and relevant clinical data have been deposited at the European Genome-Phenome Archive under the accession EGAS00001004655. The data uploaded to EGA includes all the RNA-Seq, exome, and WGS data for the exploratory, validation, and prospective cohorts from the AML PMP. Access to the data will be granted upon request and review by the BC Cancer Data Access Committee [https://www.ebi.ac.uk/ega/dacs/EGAC00000000011]. The source data underlying all Figures and Supplementary Figures are provided as a Source Data file. All the other data supporting the findings of this study are

available within the article and its supplementary information files and from the corresponding author upon reasonable request. Source data are provided with this paper.

## Code availability

Source code supporting the analysis results are available in a public repository[73] from GitHub [https://doi.org/10.5281/zenodo.4411968].

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

## Acknowledgements

This work was supported by Genome British Columbia (Grant #121AML), the BC Cancer Foundation through the Leukemia and Myeloma Program (LaMP), the BC Provincial Health Services Authority, the Terry Fox Research Institute (PPG Grant #1074), and the Leukemia and Lymphoma Society of Canada. We acknowledge Gitte Gerhard for locating specimens and extracting RNA, and thank Jennifer Grants and Kieran O'Neill for helpful discussions of the manuscript. Marion Shadbolt, Jenny Li, Ramyar Saeedi-Saghez, and Cody Lin provided informatics support. We would like to acknowledge the patients and their families who contributed to this study, and the Leukemia/Bone Marrow Transplant Program of BC for their support. The results published here are in whole or part based upon data generated by The Cancer Genome Atlas managed by the NCI and NHGRI. Information about TCGA can be found at http://cancergenome.nih.gov. The results are in part based upon data generated by the Beat AML program - a project supported by the Leukemia & Lymphoma Society and the OHSU Knight Cancer Institute. Beat AML would like to acknowledge the AML patients and Academic Medical Center partners who helped contribute samples to this program. The results published here are in whole or part based upon data generated by the Therapeutically Applicable Research to Generate Effective Treatments (TARGET) initiative, phs000218, managed by the NCI. The data used for this analysis are available at https://portal.gdc.cancer.gov/projects/TARGET-AML. Information about TARGET can be found at http://ocg.cancer.gov/programs/target. AK is supported by the John Auston BC Cancer Foundation Clinician-Scientist Award.

## Author contributions

Conceptualization, A.K.; Methodology, T.R.D., R.C., A.K.; Software, T.R.D., L.A.S., S.K.C., J.A.P., R.C., K.M.N.; Validation, T.R.D., L.A.S., S.K.C., R.C., G.D., L.C., S.M.; Formal analysis, T.R.D., J.A.P.; Investigation, T.R.D., J.D.K.P., M.J., G.D., L.C., L.A.S., S.K.C., J.A.P., R.C., K.M.N., S.M., A.M., S.M.H., J.J., X.W., R.S., K.L.M., A.J.M., R.A.M.; Resources, D.H., S.J.M.J., M.A.M., A.K.; Data curation, D.H., T.R.D., L.A.S., S.K.C.; Writing – original draft, T.R.D., J.A.P., A.K.; Writing – review & editing, T.R.D., J.D.K.P., A.K.; Visualization, T.R.D., J.A.P.; Supervision, A.J.M., R.A.M., I.B., S.J.M.J., M.A.M., D.H., A.K.; Project administration, J.D.K.P., A.K.; Funding acquisition, I.B., S.J.M.J., M.A.M., D.H., A.K.

## Competing interests

The authors declare no competing interests.

## Additional information



