## [Peer Review File · Nature Communications]

Reviewers' comments:

Reviewer #1 (Remarks to the Author):

Review

Docking et al. investigated an integrated approach for patient stratification and therapy selection in acute leukemia. This is an interesting approach. Some aspects need further clarification:

1. The abstract does not have any numbers of patients. It seems to be related to AML (title says "acute leukemia"), it does not clarify the data set. The abstract needs much more information. It is even not clear, if this data is from in silico investigations or a clinical trial.
2. This paper includes a lot of data from available data sets (TCGA, BEAT) and a huge and broad approach of bioinformatic tools. It demonstrated that thus data from exomes, genomes or RNA-Seq can be used to find out new correlations or markers in AML. However: This needs to be tested in a clinical trial, in a prospective setting, and without this, this paper is a well done performed mathematical model but cannot be translated into any clinical care.
3. This paper might be more feasible for a bioinformatically based journal demonstrating how huge data sets could be approached and combined.
4. Also any kind of speculation that some drugs can address some pathways, coming from this data set, does not help for better patient stratification or treatment if not proved by clinical data or cell lines or mouse models etc.

Reviewer #2 (Remarks to the Author):

The manuscript from Docking et al entitled "An Integrated Approach to Patient Stratification and Therapy Selection in Acute Leukemia" presents a RNA-Seq-based risk stratification model for AML based on variant-calling and gene-expression analysis. Although transcriptome-based and combined genome-transcriptome-based classification systems have been previously described, the authors demonstrate a higher accuracy of detecting all clinically relevant variants, either directly or indirectly.

Major comments:

- As stated by the authors, the sequencing data and the informatics pipelines will become publicly available. Respective links/accession numbers should be provided in the final version of the manuscript.
- For the generation of the APS gene expression signatures the authors did use only the "154 AML-like samples". How were they selected? Why were not all 171 cases with RNA-seq data used for the generation of the APS signature? Furthermore, the number of reads generated per library were very high compared to "normal" gene expression studies aiming for 60-80 million reads per transcriptome. Please comment whether such a high coverage will be needed for the variant calling? And finally it also would be nice to comment on the use of polyA enriched RNA rather than a ribosomal RNA depleted approach. Do the authors think that this might significantly influence findings?
- Throughout the manuscript it is understood that RNA-seq does not only contribute gene-expression information, but also very accurate variant-calling results. In accordance, a combined RNA-Seq based grouping into ELN 2017 risk categories and the APS signature was better than the APS signature alone. If I am not mistaken, the authors did however only compare a combined ELN-RNA-seq based classification in combination with APS versus APS alone and not a routine diagnostics based ELN classification plus APS versus APS alone. This should also be performed as addition of DNA based information might be able to further improve the model. Ultimately, one might even combine the ELN-RNA-Seq-APS classification with the conventional diagnostic information which might even further refine the model. Note: the ELN risk profile outlined in Supplementary Table S22 is not in line with the 2017 ELN risk stratification by genetics. APL and MLL-PTD are not contained in the ELN guideline.
- Based on their gene expression analyses the authors could identify pathways that are more commonly deregulated in high risk AML and which might be therapeutically exploited for further

outcome improvement. With that regard, this hypothesis was tested for the deregulation of the focal adhesion kinase (FAK) using the FAK inhibitor defactinib. While it is good that the authors have tested their hypothesis, the data provided so far are however not very strong. First, experiments seem to have been performed only in duplicates with very high defactinib concentrations in the micro-molar range, second with MDSL no real AML cell line was used, and third a marginal effect was so far only shown in one cell line model. In accordance, the authors statement "we observed that the RUNX1 knockout cell lines were preferentially sensitive to defactinib, compared to the control leukemic cell line (Fig. 8E)." is a little misleading. Additional experiments should be provided.

- Finally, the authors paper would benefit from a more advanced discussion of findings with regard to novel targeted therapies that have been recently approved for AML therapy. How can the integration of an RNA-seq based model also ensure that e.g. FLT3-ITD mutant cases are accurately assigned to a therapy including FLT3-inhibitors? And how accurately can IDH mutant patients be assigned to enasidenib/ivosidenib therapy? In line, might an RNA-seq based approach also be useful to better determine patients who might benefit from BCL2-inhibitor therapy?

Minor comments:

- In the abstract "focal adhesion kinase" is abbreviated with PTK2 instead of (FAK), which was used in the introduction as respective abbreviation. Please change and use also FAK in the abstract. One might want to add, "encoded by PTK2" to make things more clear.

- In the "Validation analysis" section the authors demonstrate that all callers seem to be more or less similar. I would suggest to make a respective statement as the GATK HaplotypeCaller is not really better and this section might be misunderstood by suggesting the other callers performed much worse.

- Through the structural variations-calling pipeline, the authors state that they were able not only to detect all the main clinically relevant gene fusions, but also to identify novel fusions not cytogenetically/molecularly previously observed. These novel fusions calls were accepted as "true" based on manual review and involvement of genes related to myeloid malignancies. However, did the authors experimentally validate these novel fusions or could they be also discovered in the WGS data? That would improve the reliability of their algorithm in order to identify rare/new translocations/fusions.

- In the Discussion section, the authors might add an additional explanation for the better performance of the RNA-seq based APS signature over the previously reported LSC17 signature. The APS signature was trained on RNA-seq data whereas the LSC17 was generated using microarray-based gene expression data. With that regard RNA-seq might be more powerful as it provides a more unbiased view.

- The Materials and Methods Section could be in large parts moved into the supplementary information.

An Integrated Approach to Patient Stratification and Therapy Selection in Acute Myeloid Leukemia - Response to Reviewers

We thank the reviewers for their positive assessment of our manuscript. In this document, we have included a point-by-point response to the Reviewers' comments. The Reviewers' comments are in quoted italics, with our responses in plain text.

Reviewer 1

"Docking et al. investigated an integrated approach for patient stratification and therapy selection in acute leukemia. This is an interesting approach. Some aspects need further clarification:"

"1. The abstract does not have any numbers of patients. It seems to be related to AML (title says "acute leukemia"), it does not clarify the data set. The abstract needs much more information. It is even not clear, if this data is from in silico investigations or a clinical trial."

We thank the reviewer for their suggestions to improve the Title and Abstract. We have updated the title of the manuscript to 'An Integrated Approach to Patient Stratification and Therapy Selection in Acute Myeloid Leukemia' to indicate that the manuscript is largely about acute myeloid leukemia. We have also made edits to the abstract to clarify the study design and include patient sample numbers.

"2. This paper includes a lot of data from available data sets (TCGA, BEAT) and a huge and broad approach of bioinformatic tools. It demonstrated that thus data from exomes, genomes or RNA-Seq can be used to find out new correlations or markers in AML. However: This needs to be tested in a clinical trial, in a prospective setting, and without this, this paper is a well done performed mathematical model but cannot be translated into any clinical care."

The major thrust of this paper is that RNA-seq as a single genomic assay can provide significantly better prognostic information than standard clinical molecular testing, and potentially predictive biomarkers to guide therapy. To directly address the reviewer's point regarding a prospective trial, we had performed RNA-seq on 42 samples from our

local cancer centre previously. We used this data to generate the APS score to predict outcome. As these assays were run over two years ago, we now have outcome data available on this prospective cohort, and can now show that the APS scores determined at diagnosis were predictive of overall survival (new Figure 3e). It should be noted that these patients were treated according to standard local regimes, independent of the APS score.

To further address this reviewer's comment, we also analyzed data from the TARGET pediatric AML cohort (156 cases), and found that the APS value provides prognostic information in pediatric AML as well (new Figure 3f). Thus we feel that the prospective cohort as well as independent validation on another large RNA-seq dataset confirms the APS as being a potentially useful adjunct to standard ELN stratification.

Further, the findings from the TARGET dataset are potentially far-reaching as they suggest that, even though the mutational spectra are somewhat different between adult and pediatric AML, similar biological pathways drive AML in children and adults. Thus the approach we undertake to examine potential therapeutic approaches later in the manuscript may have wider application than just adult AML. We have included additional text on this aspect in the Discussion (p. 19, ll. 408-412).

In summary, the APS based on our own sequencing is now validated on 3 independent AML datasets comprising adults and children with RNA-seq performed in different centres, as well as in a prospective study carried out in our institution.

"3. This paper might be more feasible for a bioinformatically based journal demonstrating how huge data sets could be approached and combined."

In this manuscript we use an array of bioinformatic analyses to devise a novel AML prognostic score that performs better than other gene expression-based classifiers, as verified in multiple datasets that include pediatric and AML patients. We also demonstrate that our analyses can help identify the potential use of novel therapies in high-risk AML patients, which we have now extensively validated. Importantly, we show that RNA-seq can be performed as a clinical-grade assay and have now validated this in a prospective trial as suggested by the Reviewer. Together, the clinical validation of an RNA-seq assay, integrated informatics analysis, and functional validation of a new prognostic score with therapeutic implication advances our understanding of the biology and potentially management of high-risk AML in a meaningful way. Thus these advances have wide general interest as they demonstrate how RNA-seq with a combination of informatics approaches can be used to improve patient stratification and potentially impact clinical management, a model that might be useful for other cancer types. We thus feel that this manuscript is best-suited for a more general audience.

"4. Also any kind of speculation that some drugs can address some pathways, coming from this data set, does not help for better patient stratification or treatment if not proved by clinical data or cell lines or mouse models etc."

To address this concern, we have undertaken further experiments to demonstrate the potential for FAK inhibition in multiple AML cell lines. Targeting of *RUNX1* using two independent shRNAs followed by treatment with either of two different FAK inhibitors that

are currently in clinical trials for solid tumors, demonstrated the sensitivity of *RUNX1*-perturbed AML cell lines to FAK inhibition. Further details of these studies are described below in our response to Reviewer 2 (Point 4), depicted in Figure 8d-f and described in the text (p. 18 ll. 368-377).

Reviewer 2

"The manuscript from Docking et al entitled "An Integrated Approach to Patient Stratification and Therapy Selection in Acute Leukemia" presents a RNA-Seq-based risk stratification model for AML based on variant-calling and gene-expression analysis. Although transcriptome-based and combined genome-transcriptome-based classification systems have been previously described, the authors demonstrate a higher accuracy of detecting all clinically relevant variants, either directly or indirectly."

"Major comments:"

1. "As stated by the authors, the sequencing data and the informatics pipelines will become publicly available. Respective links/accession numbers should be provided in the final version of the manuscript."

The data underlying the manuscript has been submitted to EGA (Accession identifier EGAS00001004655). Data used in this manuscript has been submitted as Source Data to Nature and, custom code underlying the analysis is available on GitHub (<https://github.com/rdocking/amlpmpsupport>).

2. "For the generation of the APS gene expression signatures the authors did use only the "154 AML-like samples". How were they selected? Why were not all 171 cases with RNA-seq data used for the generation of the APS signature? Furthermore, the number of reads generated per library were very high compared to "normal" gene expression studies aiming for 60-80 million reads per transcriptome. Please comment whether such a high coverage will be needed for the variant calling? And finally it also would be nice to comment on the use of polyA enriched RNA rather than a ribosomal RNA depleted approach. Do the authors think that this might significantly influence findings?"

We only considered the 154 AML-like cases for generation of the APS gene expression signature, and excluded the remaining samples from MDS cases. In particular, inclusion of MDS samples in training the signature would be expected to limit the signature's performance on AML cases. We did retain the MDS cases for the initial variant-calling exercise, since SNV and fusion variant detection are expected to be similar for AML and MDS cases, and we were interested in the proof-of-principle of using an RNA-seq strategy for variant calling across myeloid malignancies.

Regarding the depth of sequencing, we have re-examined the empirical coverage distributions observed for RNA-Seq libraries sequenced with different numbers of reads. Given that the mean coverage depth at variant sites was over 300x in RNA-Seq (Cohort 2) (see Table S4, Fig. S2-S4), which was sequenced to ~200 million reads, we believe lower sequencing depths of 60-80 million reads would result in sufficient coverage (~100x) for variant calling and fusion detection based on minimum coverage requirements for the

clinical myeloid panel at our institution. We have added a note in the text to this effect (p. 7, l. 119-123).

To determine whether Ribo-depleted library generation would perform comparably to polyA RNA-Seq libraries to determine the APS score, we compared matched RNA-Seq libraries generated using a Ribo-depletion-based library protocol to PolyA-enriched libraries from the same samples. The data show that, for Ribo-depleted libraries with good mapping rates (>40%), the correlation r^2 between matched polyA and Ribo-depleted libraries is >0.89. This indicates that the APS value would be translatable to alternate library construction methods, if an appropriate correction factor is applied. These data have been included in the paper as Supplemental Figure S11 (cited on p. 11, l. 223-226).

3. "Throughout the manuscript it is understood that RNA-seq does not only contribute gene-expression information, but also very accurate variant-calling results. In accordance, a combined RNA-Seq based grouping into ELN 2017 risk categories and the APS signature was better than the APS signature alone. If I am not mistaken, the authors did however only compare a combined ELN-RNA-seq based classification in combination with APS versus APS alone and not a routine diagnostics based ELN classification plus APS versus APS alone. This should also be performed as addition of DNA based information might be able to further improve the model. Ultimately, one might even combine the ELN-RNA-Seq-APS classification with the conventional diagnostic information which might even further refine the model. Note: the ELN risk profile outlined in Supplementary Table S22 is not in line with the 2017 ELN risk stratification by genetics. APL and MLL-PTD are not contained in the ELN guideline."

To address this point, it is important to note that the routine diagnostic information for the AML PMP data set only included cytogenetics, as well as *FLT3*-ITD and *NPM1* status for most patients. Therefore, the only additional clinical diagnostic information available is the patient karyotype. We have added to the paper additional stratification models, making use of the diagnostic karyotype information in addition to mutation calls from the RNA-Seq assay, in order to approximate the stratifications that would be expected for a modern routine diagnostics-based AML assay. To determine how incorporation of diagnostic karyotyping would alter the ELN-RNA-Seq-APS stratification, we generated an additional stratification model (incorporating the APS signature together with the cytogenetic-based stratification), and re-reviewed those stratifications to identify cases that would be affected, and discuss those cases in the text (pp. 15 ll. 303-317, Figure S15). Though the karyotype-based and RNA-Seq-based models incorporating APS stratify most patients to the same categories, we observed that the incorporation of the karyotype information would not necessarily improve upon the RNA-Seq based stratifications (though a much larger patient cohort would be required to test this).

Further, we thank the reviewer for raising the important point that APL and *MLL*-PTD are not included in the ELN guidelines. As ELN guidelines are not static, and we were interested in including known strong influencers of risk to truly understand the value of the APS, we included APL and *MLL*-PTD (Patel et al. 2012, PMID: 22417203). We have now noted this in the manuscript (pp. 36, ll. 779-783), and provided an additional supplemental table more clearly documenting the stratification rules used (Table S22).

4. *“Based on their gene expression analyses the authors could identify pathways that are more commonly deregulated in high risk AML and which might be therapeutically exploited for further outcome improvement. With that regard, this hypothesis was tested for the deregulation of the focal adhesion kinase (FAK) using the FAK inhibitor defactinib. While it is good that the authors have tested their hypothesis, the data provided so far are however not very strong. First, experiments seem to have been performed only in duplicates with very high defactinib concentrations in the micro-molar range, second with MDSL no real AML cell line was used, and third a marginal effect was so far only shown in one cell line model. In accordance, the authors statement “we observed that the RUNX1 knockout cell lines were preferentially sensitive to defactinib, compared to the control leukemic cell line (Fig. 8E).” is a little misleading. Additional experiments should be provided.”*

We agree with the Reviewer’s comment on the need for additional experimentation to confirm our data. We have now performed a substantial set of additional experiments. First, to augment the MDSL results, we repeated all experiments in triplicate, added a second FAK inhibitor (GSK2252098), and examined results in *TP53*-KO models as well as *RUNX1*-KO models. The data confirm our previous assessment. To accommodate new experiments in AML cell lines in the main body of the paper as requested by the Reviewer (see below), the panel showing these data has been moved to a Supplemental Figure (Figure S18, p. 18, l. 368-372).

To address the concern that ‘no real AML cell line was used’, we performed experiments in two AML cell lines: KG-1a and THP-1. In both of these cell lines, we used two independent shRNAs targeting different regions to knock down *RUNX1* expression, and treated cells with two distinct FAK inhibitors: VS-4718 and Defactinib. For both cell lines targeting of *RUNX1* resulted in significantly fewer colonies in colony-forming cell assays when cells were treated with FAK inhibitors compared to vehicle control (Figure 8e,f, p. 18, l. 372-377). Additionally, we observed induction of FAK in both cell lines upon *RUNX1* knockdown (by Western blots) (Figure 8d).

With these additional experiments, we feel that our assertion that *RUNX1*-targeted cell lines are preferentially sensitive to FAK inhibition is significantly strengthened.

5. *“Finally, the authors paper would benefit from a more advanced discussion of findings with regard to novel targeted therapies that have been recently approved for AML therapy. How can the integration of an RNA-seq based model also ensure that e.g. FLT3-ITD mutant cases are accurately assigned to a therapy including FLT3-inhibitors? And how accurately can IDH mutant patients be assigned to enasidenib/ivosidenib therapy? In line, might an RNA-seq based approach also be useful to better determine patients who might benefit from BCL2-inhibitor therapy?”*

The intent of the RNA-Seq assay is for it to be incorporated within clinical laboratories preparing reports with the assistance of trained clinical geneticists. The intent of the large effort put into assessing and validating SNV-calling performance was largely undertaken precisely to ensure that *FLT3*-ITD mutant and *IDH*-mutant cases can be assigned to appropriate targeted therapies. Indeed, one interesting future research direction is to use

the RNA-Seq assay in patient cohorts being treated with FLT3 inhibitors, to better understand how factors like ITD length and expression level bear on patient response to those drugs. We have updated the Discussion text to note these points (p. 19, ll. 397-403).

At the Reviewer's suggestion, we also explored whether RNA-seq could identify patients more likely to respond to a BCL2 inhibitor. The BEAT AML data include Venetoclax as a single agent in their chemogenomic screen. Our findings show that AML samples with higher APS values are slightly more likely to respond to Venetoclax, though the relationship is not statistically significant. This is most likely due to the variation within high-APS AMLs. We believe that an RNA-Seq assay could be used to identify predictors of Venetoclax response, but we considered that question outside the scope of the current paper.

"Minor comments:"

"In the abstract "focal adhesion kinase" is abbreviated with PTK2 instead of (FAK), which was used in the introduction as respective abbreviation. Please change and use also FAK in the abstract. One might want to add, "encoded by PTK2" to make things more clear."

We thank the Reviewer for this suggestion, and have adjusted the manuscript as recommended.

"In the "Validation analysis" section the authors demonstrate that all callers seem to be more or less similar. I would suggest to make a respective statement as the GATK HaplotypeCaller is not really better and this section might be misunderstood by suggesting the other callers performed much worse"

This is an important point to clarify, and we thank the Reviewer for noting this. We have modified the text on p. 9, ll. 164-166 as suggested.

"Through the structural variations-calling pipeline, the authors state that they were able not only to detect all the main clinically relevant gene fusions, but also to identify novel fusions not cytogenetically/molecularly previously observed. These novel fusions calls were accepted as "true" based on manual review and involvement of genes related to myeloid malignancies. However, did the authors experimentally validate these novel fusions or could they be also discovered in the WGS data? That would improve the reliability of their algorithm in order to identify rare/new translocations/fusions."

To improve the reliability of the fusion detection algorithm, we undertook several approaches to validate the novel fusion calls. In all cases, we manually reviewed the RNA-Seq data in IGV to identify reads spanning the putative fusion breakpoints (22 cases). Additionally, for the 19 cases where copy-number array data was available, we attempted to validate the gain or loss of signal in regions indicated as being involved in the fusion. In one other case, we were able to use the WGS data to confirm the call. This verification effort identified two sets of fusion partners which we were unable to verify (t(5;12) _IL31RA_- _CTDSP2_ and t(X;Y) _CD99P1_- _CD99). Unfortunately, additional patient material was not available for validation in these two cases. We have added text to note this verification in the Methods (p. 33, ll. 700-707), and updated Table S15 with the verification results.

“In the Discussion section, the authors might add an additional explanation for the better performance of the RNA-seq based APS signature over the previously reported LSC17 signature. The APS signature was trained on RNA-seq data whereas the LSC17 was generated using microarray-based gene expression data. With that regard RNA-seq might be more powerful as it provides a more unbiased view.”

We thank the Reviewer for this suggestion and have added this point to the Discussion (p. 20, ll. 421-422).

“The Materials and Methods Section could be in large parts moved into the supplementary information.”

As requested by the Editor, we have left the Methods included in the main text.

REVIEWERS' COMMENTS

Reviewer #1 (Remarks to the Author):

no more comments

Reviewer #2 (Remarks to the Author):

In the revised version of the manuscript the authors have addressed all major concerns, especially that the data and pipelines will become publicly available, that the cell line data become more solid and that additional test cohorts are included.